# The AP-2 complex interacts with γ-TuRC and regulates the proliferative capacity of neural progenitors

Santiago Camblor-Perujo[1],*, Ebru Ozer Yildiz[1],*, Hanna Küpper[1], Melina Overhoff[1,2], Saumya Rastogi[1], Hisham Bazzi[1,3,4], Natalia L Kononenko[1,2,3,5]

**Centrosomes are organelles that nucleate microtubules via the activity of gamma–tubulin ring complexes (γ-TuRC). In the developing brain, centrosome integrity is central to the progression of the neural progenitor cell cycle, and its loss leads to microcephaly. We show that NPCs maintain centrosome integrity via the endocytic adaptor protein complex-2 (AP-2). NPCs lacking AP-2 exhibit defects in centrosome formation and mitotic progression, accompanied by DNA damage and accumulation of p53. This function of AP-2 in regulating the proliferative capacity of NPCs is independent of its role in clathrin-mediated endocytosis and is coupled to its association with the GCP2, GCP3, and GCP4 components of γ-TuRC. We find that AP-2 maintains γ-TuRC organization and regulates centrosome function at the level of MT nucleation. Taken together, our data reveal a novel, noncanonical function of AP-2 in regulating the proliferative capacity of NPCs and open new avenues for the identification of novel therapeutic strategies for the treatment of neurodevelopmental and neurodegenerative disorders with AP-2 complex dysfunction.**

## Introduction

The number of neurons generated in the brain neocortex is determined by the proliferative capacity of a relatively small number of neural progenitor cells (NPCs) (Noctor et al, 2004; Gotz & Huttner, 2005). The proliferative capacity of NPCs is determined by cell-intrinsic factors that dictate the number of their division rounds, and by the varying identity of the daughter cells generated during division (Salomoni & Calegari, 2010; Franco & Müller, 2013; Xing et al, 2021). Aberrations in NPC proliferation lead to diseases ranging from neurodevelopmental disorders to cancer, and the maintenance of their proliferative capacity is the subject of extensive investigations (Homem et al, 2015).

NPC proliferation is coupled to cell cycle regulation, and differences in cycle length or mitotic spindle formation and/or positioning affect the probability of promoting proliferation or differentiation into neuroblasts (Calegari & Huttner, 2003; Calegari et al, 2005; Lange et al, 2009; Pilaz et al, 2016). A key role in the formation of the bipolar mitotic spindle is governed by the centrosome (Strome et al, 2001; O'Toole et al, 2012), which is typically a non-membrane-bound organelle, although its encirclement by an endoplasmic reticulum-derived membrane has recently been reported (Maheshwari et al, 2023). Centrosomes consist of a pair of centrioles and a surrounding matrix and function as the major microtubule (MT)-organizing centers (MTOC) in animal cells (Luders & Stearns, 2007; Petry & Vale, 2015; Lin et al, 2022). In this process, the γ-tubulin ring complex (γ-TuRC), which consists of γ-tubulin, GCP2, GCP3, and at least three other proteins (GCP4, GCP5, and GCP6) (Wieczorek et al, 2020; Zupa et al, 2021), is anchored to the centrosome by interaction with pericentrin, among others, and serves as a template for $\alpha/\beta$ tubulin dimers to form the growing MT (Schiebel, 2000; Liu et al, 2020). Nucleation from MTOCs requires not only recruitment but also specific activation of γTuRC, and at interphase centrosomes, both functions can be controlled by the microtubule polymerase CKAP5 (ch-TOG) (Thawani & Petry, 2021; Ali et al, 2023). Furthermore, during spindle assembly, ch-TOG may also cooperate with γTuRC in chromatin or augmin-mediated nucleation (Roostalu et al, 2015). Consistent with this important function of centrosomes in mitotic spindle formation, centrosome integrity is a crucial regulator of proliferative potential in mammalian cells. Cells that either lack centrosomes or have disrupted centrosomes undergo irreversible G1 cell cycle arrest or cell death (Hinchcliffe et al, 2001; Khodjakov & Rieder, 2001; Mikule et al, 2007; Bazzi & Anderson, 2014; Insolera et al, 2014; Lambrus et al, 2015; Wong et al, 2015). Human mutations in genes encoding centrosomal proteins cause microcephaly or a small brain (Bond et al, 2005). Mouse models lacking functional centrosomes are also characterized by microcephaly because of the depletion of NPCs by apoptosis and early onset of neuronal

[1]CECAD Excellence Center, University of Cologne, Cologne, Germany [2]Center for Physiology, Faculty of Medicine and University Hospital Cologne, University of Cologne, Cologne, Germany [3]Center for Molecular Medicine Cologne, Faculty of Medicine and University Hospital Cologne, University of Cologne, Cologne, Germany [4]Department of Dermatology and Venereology, Faculty of Medicine and University Hospital Cologne, University of Cologne, Cologne, Germany [5]Institute of Genetics, Natural Faculty, University of Cologne, Cologne, Germany

Correspondence: n.kononenko@uni-koeln.de
*Santiago Camblor-Perujo and Ebru Ozer Yildiz contributed equally to this work

differentiation (Insolera et al, 2014; Marjanović et al, 2015). Importantly, the microcephaly phenotype can be rescued by the additional deletion of p53, which bypasses the cell death pathway and allows NPC proliferation and expansion upon the loss of centrosomes (Insolera et al, 2014).

The balance between cell proliferation and differentiation is maintained by a number of mechanisms, including clathrin-mediated endocytosis (CME) (Royle, 2012; Narayana et al, 2019). CME constitutes the main entry route for most surface receptors and their ligands in various cell types (Reider & Wendland, 2011) and is initiated by the assembly of a clathrin-coated pit on the plasma membrane (Kirchhausen, 2000). Clathrin recruitment to the plasma membrane requires the action of cargo adaptors, including assembly protein (AP) complex 2 (AP-2) (Conner & Schmid, 2003; Schmid & McMahon, 2007). AP-2 is the major clathrin adaptor protein for the CME (Robinson, 2004), which functions as a heterotetrameric complex comprised of $\alpha$, $\beta$, $\mu$, and $\sigma$ subunits (Dittman & Ryan, 2009). Brain-specific functions of CME include activity-dependent recycling of synaptic vesicles (Kononenko & Haucke, 2015) and regulation of cell differentiation during neurodevelopment (Camblor-Perujo & Kononenko, 2022). The latter involves control of asymmetrical partitioning of cell determinants into two daughter cells through CME-dependent internalization of receptors and ligands localized in the plasma membrane (Uemura et al, 1989; Berdnik et al, 2002; Nishimura & Kaibuchi, 2007; Zhou et al, 2011; Nahorski et al, 2015, 2018).

In contrast to the well-known function of CME in neuronal differentiation, we know much less about the direct role of CME components in NPC proliferation. In fact, most of the research to date has been conducted in nonneuronal cells, where the activity of endocytic processes was initially suggested to be inhibited during mitosis (Pypaert et al, 1991; Warren, 1993; Raucher & Sheetz, 1999), although later studies have challenged this view (Boucrot & Kirchhausen, 2007; Tacheva-Grigorova et al, 2013). In nonneuronal cells, CME components control mitotic progression independently of their canonical roles in endocytosis and localize to the centrosome, mitotic spindle or midbody to regulate the spindle morphology and/or cytokinesis (Thompson et al, 2004; Royle et al, 2005; Lehtonen et al, 2008; Booth et al, 2011; Olszewski et al, 2014). Despite these numerous studies in nonneuronal cells, the precise role of the CME, and in particular, its key adaptor AP-2, in controlling neuronal progenitor proliferation has not been explored in detail.

Here, we show that AP-2 is required in NPCs to maintain centrosome integrity. NPCs lacking AP-2 are deficient in centrosome function and exhibit mitotic delays accompanied by cell cycle progression defects and apoptosis. We find that this function of AP-2 in regulating the proliferative capacity of NPCs is independent of its canonical role in CME and is coupled to its association with GCP2, GCP3, and GCP4 components of $\gamma$-TuRC. AP-2 regulates the localization of $\gamma$-TuRC components to the centrosome and its loss impairs centrosomal MT growth in NPCs, and in MEFs. Taken together, our data identify a novel, noncanonical function of the endocytic adaptor AP-2 in regulating the proliferative capacity of mammalian cells, including NPCs and MEFs.

# Results

## AP-2 regulates cell cycle progression of NPCs and this function is independent of the role of clathrin in CME

To directly test the role of AP-2 in NPC proliferation, we used a mouse line in which the deletion of AP-2 is driven by tamoxifen (Tmx)-dependent recombination of the central $\mu$-subunit (*Ap2m1* loxP: CAG-Cre$^{Tmx}$) (Kononenko et al, 2017). Deletion of the $\mu$-subunit destabilizes the AP-2 complex, leading to subsequent degradation of its $\alpha$-subunit (Kononenko et al, 2014). We isolated primary NPCs from telencephalic ventricles of *Ap2m1* loxP: CAG-Cre$^{Tmx}$ mice at embryonic stage 12.5 (E12.5, peak of NPC proliferation) and observed a strong down-regulation of the AP-2 complex monitored via its AP-2$\alpha$ subunit in NPCs treated with Tmx for 72 h (cultured and treated with Tmx for 3 d, hereafter defined as days in vitro 3, DIV3) (Fig 1A and B). Interestingly, this short-term AP-2 deletion resulted in a large reduction in cell number of NPCs treated with Tmx (defined as KO) compared with NPCs treated with ethanol (defined as WT) (Fig 1C). The reduction in cell number was even more dramatic in KO NPCs cultured for 7 d (Fig S1A). Live imaging of WT and KO NPCs at DIV3 revealed an inability of KO neurospheres to grow (Fig 1D). This growth defect of AP-2 KO NPCs was also accompanied by drastic morphological alterations from the typical round shape in WT to an elongated tubular shape, a feature usually characteristic of terminally differentiated cells (Figs 1E and S1B). Moreover, only about half of the NPCs lacking AP-2 incorporated EdU during a 24-h pulse (Fig 1F and G), providing evidence that AP-2 may regulate NPC proliferation and progression through the cell cycle.

To gain direct insights into the cellular and molecular mechanisms that are dysregulated in NPCs lacking AP-2, we next performed a global proteome analysis of WT and AP-2 KO NPCs at DIV3. Mass spectrometry (MS) analysis identified 22 significantly up-regulated proteins (q < 0.05) and 146 significantly down-regulated proteins (q < 0.05) in NPCs lacking AP-2. Among significantly down-regulated proteins, we identified AP-2$\mu$ (AP2M1), confirming the efficiency of AP-2 KO in our model (Fig 1H). In agreement with the data presented in Fig 1A and B, the levels of AP-2$\alpha$ (AP2A1) were also reduced. In line with this, uptake of transferrin, known to enter cells via CME, was decreased by approximately 80% in AP-2 KO NPCs (Fig S1C and D). Interestingly, this short-term AP-2$\mu$ deletion had no effect on the total levels of clathrin and/or other CME proteins (Table S1). Whereas the up-regulated proteins could not be assigned to any specific pathway (Fig S1E and F), the proteins down-regulated in the absence of AP-2 were mainly enriched in pathways controlling cell cycle and RNA processing (Figs 1I and S1G). We further found that most of the down-regulated proteins were under the control of key cell cycle kinases, including CHEK1, CDK2, and CDK1 (Fig 1J). In addition, analysis performed using the Mammalian Phenotype Ontology Database showed that proteins down-regulated upon deletion of AP-2 were associated with embryonic lethality before implantation, akin to the very early embryonic lethality (before day 3.5 post-coitus) of mice constitutively lacking AP-2$\mu$ (Mitsunari et al, 2005) (Fig 1K). Flow cytometric quantification of DNA content using DNA-binding propidium iodide demonstrated that AP-2 is required for cell cycle progression, as significantly less DNA was observed in the S and G2/M

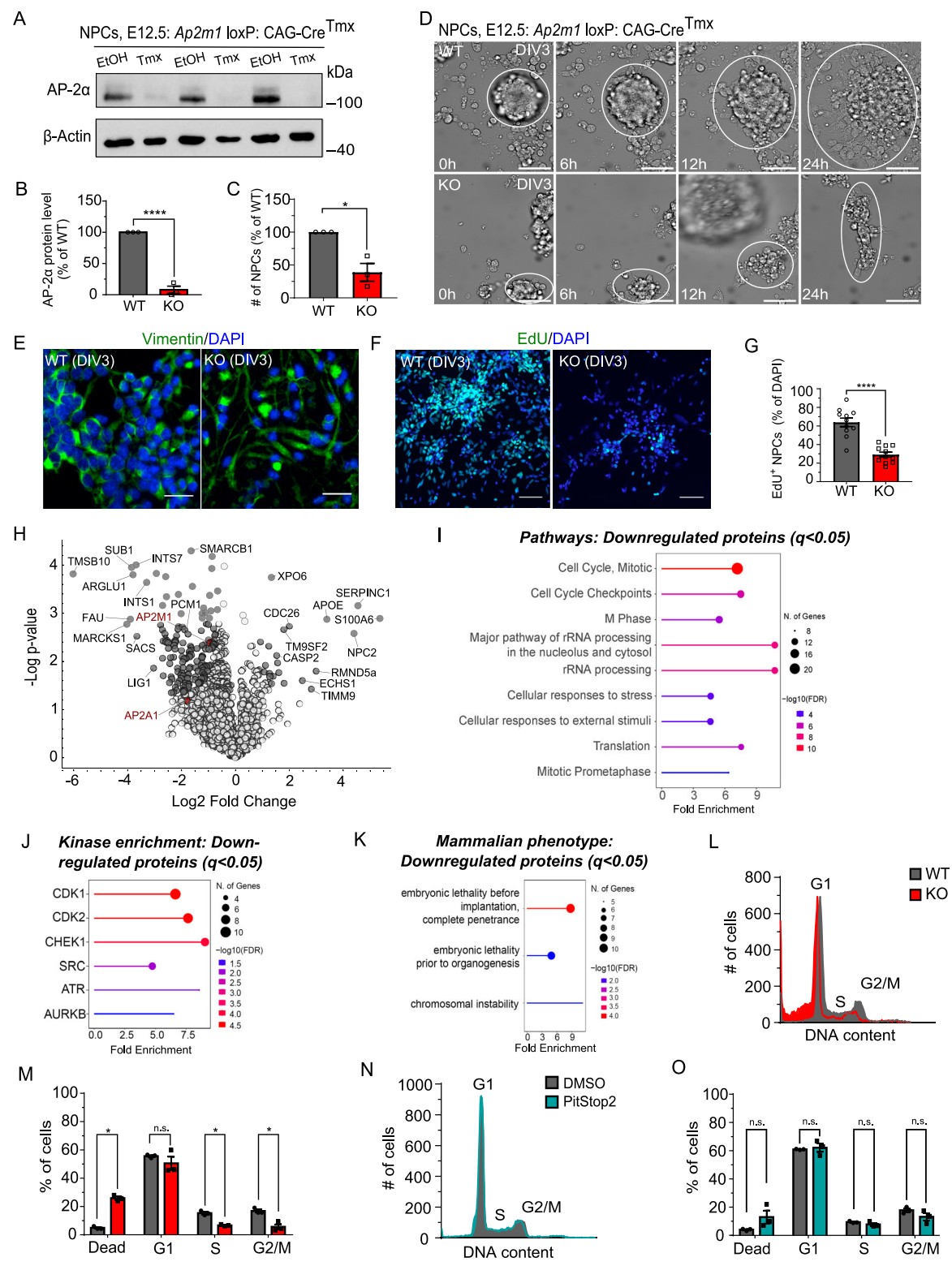

**Figure 1. AP-2 regulates neuronal cell cycle progression independent of clathrin.**
**(A, B)** Immunoblot analysis of AP-2α protein levels in lysates of NPCs isolated from *Ap2m1* loxP: CAG-Cre-ER$^{Tmx}$ mice at E12.5 and cultured for 3 d (DIV3). NPCs were treated at DIV1 with either tamoxifen to induce KO (Tmx) or ethanol (EtOH) to be used as a WT (WT set to 100%, KO: 7.88% ± 5.84%, $P < 0.0001$, $N_{WT}$ = 3, $N_{KO}$ = 3, two-tailed unpaired $t$ test). **(C)** Relative number of WT and AP-2 KO NPCs at DIV3 (WT set to 100%, KO = 38.80 ± 13.50, $P$ = 0.01, $N_{WT}$ = 3, $N_{KO}$ = 3, two-tailed unpaired $t$ test). **(D)** Timestamps of 24-h live imaging of WT and KO NPCs at DIV3 showing an increase in neurosphere growth (white circles) in WT but not in KO. Representative examples from $N_{WT}$ = 3, $N_{KO}$ = 3. Scale bar, 50 μm. **(E)** WT and KO NPCs at DIV3, immunostained for vimentin. DAPI stain was used to visualize the nuclei. In the KO condition, lower cell number and

stages of the cell cycle in NPCs lacking AP-2 compared with controls (Fig 1L). Consistent with the finding that caspase-2 is up-regulated in KO NPCs (see Fig 1H), we observed that a fraction of AP-2 KO cells had reduced DNA content, likely reflecting cell death (Fig 1L and M).

To understand whether the observed defects in cell cycle progression in AP-2 KO NPCs described above are the result of dysfunctional CME, we treated primary NPCs with selective membrane-permeable inhibitor of clathrin terminal domain Pitstop2. Pitstop2 inhibits the association of clathrin with clathrin box motif-containing CME proteins but does not affect AP-2 function (von Kleist et al, 2011). We have previously shown that Pitstop2 used at a concentration of 30 μM effectively arrests CME in primary neurons (Kononenko et al, 2014). Furthermore, this concentration has been shown to be nontoxic in several cell lines even after prolonged (24 h and longer) exposure (von Kleist et al, 2011; Smith et al, 2013). In primary NPCs treated with 30 μM of Pitstop2 for 24 h, we observed a significant down-regulation of transferrin uptake (Fig S1H and I). Importantly, this down-regulation of about 60–80% was comparable with endocytic phenotype AP-2 KO NPCs (see Fig S1D). Thus, we analyzed the DNA content of NPCs treated with 30 μM of Pitstop2 at DIV2 for 24 h (to mimic the effect of AP-2 KO) or NPCs treated with DMSO by flow cytometry. In contrast to deletion of AP-2, inhibition of clathrin function in CME had no effect on DNA content (Fig 1N and O). Because CME was previously shown to regulate cell progression through the G1 restriction point by controlling internalization of EGF-bound receptors and downstream mitogenic ERK signaling (Sigismund et al, 2008), we also analyzed ERK signaling by Western blotting and immunocytochemistry in AP-2 KO NPCs. We found that AP-2 deletion resulted in increased ERK activation (Fig S1J and K), suggesting that the defects in cell cycle progression are not because of diminished mitogenic ERK signaling. Taken together, these data suggest that AP-2 regulates cell cycle progression in primary dividing NPC cells independently of CME and clathrin.

### AP-2 associates with components of γ-TuRC and its loss impairs centrosome organization

To reveal the precise cellular and molecular mechanism behind AP-2-mediated regulation of neuronal progenitor proliferation,

we performed mass spectrometry analysis of AP-2-binding partners in E12.5 embryonic cortex by using an AP-2α antibody as bait. The specificity of the AP-2α antibody has been confirmed using AP-2 KO NPC lysates (see Fig 1A). In addition to AP-2 complex components and known AP-2 interactors such as clathrin, AP180, and HIP1, we identified the γ-tubulin (TUBG1) and γ-TuRC components GCP2, GCP3, GCP4, and GCP8 (also known as MZT2) as novel putative interaction partners of the AP-2 complex in the developing cortex (Fig 2A and Table S2). γ-TuRC is an essential component of centrosomes and other MTOCs. Consequently, we speculated that AP-2 controls cell proliferation by facilitating γ-TuRC-dependent MT nucleation from the centrosome. To explore this hypothesis, we first analyzed the AP-2/γ-TuRC interaction using co-immunoprecipitation assays and colocalization studies. We found that in the cortex of E12-14 mouse embryos, endogenous AP-2α forms a complex and co-immunoprecipitates with γ-tubulin, GCP4 and GCP3 components of γ-TuRC (Fig 2B). This complex contained AP-2μ (Fig S2A) and the association of AP-2α with GCP2 and GCP3 was also observed in lysates from the cortex of 8-wk-old mice (Fig 2C). Conversely, AP-2μ and AP-2α were found in complex with GCP2 (Fig S2B) and GCP3 (Fig S2C), albeit with lower efficiency. Colocalization studies in NPCs revealed that AP-2α and AP-2μ associate with GCP3 at the centrosome in interphase and ana-/telophase during mitosis (Figs 2D and E and S2D), whereas the loss of AP-2 in these cells resulted in a significant decrease and delocalization of GCP3 and GCP2 levels at the centrosome (marked by either centriolar marker CEP135 or by γ-tubulin) but not of γ-tubulin per se (Figs 2F–H and S2E–I). In addition, the prominent PCM protein PCNT was also disorganized in NPCs lacking AP-2 (Fig 2I and J). These changes were accompanied by a decrease in the levels of the centrosomal microsatellite marker PCM1 (Fig 2K and L), in line with its decreased levels in MS analysis of AP-2 KO NPCs (see Fig 1H). Moreover, whereas more than 80% of WT NPCs were characterized by the presence of a single centrosome visualized by the immunostaining with γ-Tubulin, AP-2μ-deficient NPCs contained either no centrosomes (>60%) or multiple centrosomes (about 20%) (Fig 2M and N). These multiple centrosomes in AP-2 KO NPCs also contained the bona fide marker for centrioles, CEP135,

---

elongated tubular morphology are observed. Representative fluorescent images from $N_{WT}$ = 3, $N_{KO}$ = 3. Scale bars, 25 μm. **(F)** Proliferation ability of WT and AP-2 NPCs at DIV3 analyzed using the EdU pulse assay, where EdU is directly coupled with Alexa488. DAPI stain was used to visualize the nuclei. Scale bars: 50 μm. **(G)** Percentage of proliferating WT and KO NPCs cells (EdU⁺ cells) (WT = 63.93% ± 4.56%, KO = 29.10% ± 2.73%, $P$ < 0.0001, two-tailed unpaired $t$ test). In total, 11 WT/KO images, with at least 300 EdU cells and 1,500 total cells per condition, from N = 3. **(H)** Volcano plot of differentially expressed proteins in WT and AP-2 KO NPCs at DIV3 analyzed using label-free proteomic analysis (N = 3 WT/KO). Dark gray circles indicate all protein deregulated at q < 0.05. AP2M1 and AP2A1 subunits of the AP-2 complex are highlighted in red. **(I)** ShinyGO v0.77-based GO analysis of pathways ("Reactome pathways") in the down-regulated proteome of AP-2 KO NPCs compared with the WT (cut-off q < 0.05). **(J)** ShinyGO v0.77-based kinase enrichment analysis in the down-regulated proteome of AP-2 KO NPCs compared with the WT (cut-off q < 0.05). **(K)** ShinyGO v0.77-based mammalian phenotype analysis in the down-regulated proteome of AP-2 KO NPCs compared with the WT (cut-off q < 0.05). **(L, M)** Cell cycle analysis by DNA content estimation with flow cytometry in WT and AP-2 NPCs at DIV2. Histogram in (M) indicates the percentages of dead cells or cells in G1, S, and G2/M phases of cell cycle ($WT^{DeadCells}$ = 4.88% ± 0.57%, $KO^{DeadCells}$ = 25.89% ± 1.21%, $P$ < 0.016, $WT^{G1}$ = 55.58% ± 0.68%, $KO^{G1}$ = 50.46% ± 4.68%, $P$ = 0.339, $WT^{S}$ = 15.28% ± 0.88%, $KO^{S}$ = 6.66% ± 0.39%, $P$ = 0.031, $WT^{G2/M}$ = 16.83% ± 1.10%, $KO^{G2/M}$ = 5.60% ± 1.89%, $P$ = 0.0168, two-way ANOVA with Holm-Sidak's multiple comparisons test). In addition, significant differences were found between WT and KO in the overall distribution of cell cycle phases with the $x^2$ test using the original cell counts ($P$ < 0.001 for each independent N). WT = 20,000 events and KO ≥ 5500 events per each experiment, from N = 3. **(N, O)** Cell cycle analysis by DNA content estimation with flow cytometry in NPCs treated for 24 h either with DMSO as a control or with 30 μM of Pitstop2 and analyzed at DIV2. Histogram in (H) indicates the percentages of dead cells or cells in G1, S, and G2/M phases of the cell cycle ($DMSO^{DeadCells}$ = 5.73% ± 1.99%, $Pitstop2^{DeadCells}$ = 11.22% ± 5.50%, $DMSO^{G1}$ = 60.89% ± 0.19%, $Pitstop2^{G1}$ = 62.10% ± 3.00%, $DMSO^{S}$ = 9.37% ± 0.38%, $Pitstop2^{S}$ = 7.74% ± 1.01%, $DMSO^{G2/M}$ = 17.77% ± 1.27%, $Pitstop2^{G2/M}$ = 13.11% ± 2.59%, two-way ANOVA with Holm–Sidak's multiple comparisons test). In addition, no significant differences were found between DMSO and Pitstop2-treated condition in the overall distribution of cell cycle phases with the $x^2$ test using the original cell counts. In total, 20,000 events per condition, from N = 3. Data information: all graphs show mean ± SEM. n.s.—nonsignificant. * indicates $P$ ≤ 0.05; ** indicates $P$ ≤ 0.01; *** indicates $P$ ≤ 0.001; **** indicates $P$ ≤ 0.0001. N represents independent cultures obtained from independent animals.

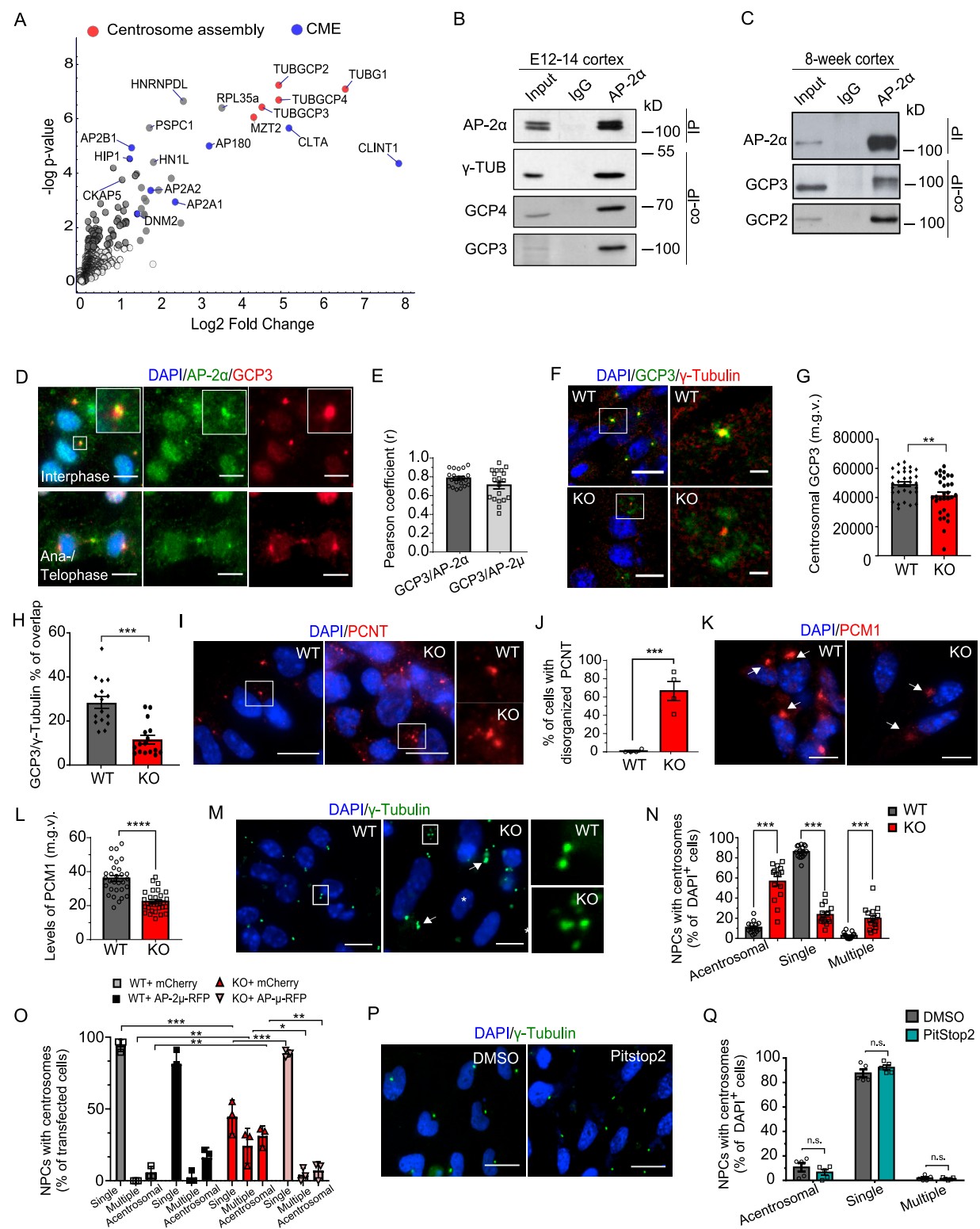

**Figure 2. AP-2 associates with components of γ-TuRC and its loss impairs centrosome organization.**
**(A)** Volcano plot showing interaction partners identified by pull-down of AP-2α from the cortex of E12.5 mice followed by quantitative mass spectrometry. Proteins significantly enriched in the AP-2α pull-down compared with the IgG control are highlighted in gray. AP-2 interactors with the function in endocytosis/centrosome assembly are highlighted in blue/red, accordingly. N = 5. **(B)** Co-immunoprecipitation of endogenous AP-2α with γ-Tubulin, GCP4 and GCP3 in E12.5 cortex lysates. Input, 1.5% of the lysate. Representative example from N = 3. **(C)** Co-immunoprecipitation of endogenous AP-2α with GCP2 and GCP3 in 8-wk-old cortex lysates. Input, 1% of the lysate. Representative example from N = 3. **(D)** NPCs at DIV3 immunostained for AP-2α with GCP3 using methanol-based fixation. DAPI stain was used to visualize the

whereas approximately 30% of acentrosmal NPCs lacked both centrosomal markers (Fig S2J and K), a phenotype which was rescued by overexpressing AP-2µ in these cells (Figs 2O and S2L). Consistent with the CME-independent role of AP-2 in cell cycle regulation shown above, centrosome integrity was not altered in NPCs treated with Pitstop2 (Fig 2P and Q) or in NPCs in which clathrin function was inhibited by shRNA-mediated knockdown of the clathrin heavy chain (CHC) (Fig S2M–R). Overall, the data show that neuronal AP-2 associates with γ-TuRC components and that loss of AP-2 results in disorganization of centrosome components.

### AP-2 regulates the function of the centrosome at the level of MT nucleation

Although it is known that AP-2 is canonically required at the PM to regulate the CME, we and others have previously identified additional roles for AP-2 in mediating MT-based intracellular transport of various cargoes, including TrkB (Kononenko et al, 2017; Andres-Alonso et al, 2019) and BACE1 (Bera et al, 2020). Thus, we speculated that the association of AP-2 with γ-TuRC components described above might be a result of its recruitment to the centrosome. To test this, we analyzed the intracellular trafficking of the mCherry-tagged AP-2µ towards the centrosome, marked by GFP-tagged γ-tubulin in NPCs at DIV3 (Fig 3A and B and Video 1). We observed that ~50% of all AP-2µ moving puncta trafficked towards the centrosome (Fig 3C). MT polymerization was not altered in AP-2 KO NPCs, because they were able to regrow MTs after their disassembly by nocodazole (Fig 3D and E), suggesting that AP-2 is not per se required for MT nucleation. However, we observed that whereas WT NPCs grew new MTs in a pattern originating from the centrosome, NPCs lacking AP-2 generated MTs with no clear centrosomal origin (Fig 3D and F).

Furthermore, the number of centrosomal MTs was also significantly decreased in these cells (Fig 3G and H). These results suggest that AP-2 might be required for centrosome integrity and to grow MTs specifically at the centrosome. To explore this hypothesis, we analyzed centrosomal (marked by γ-tubulin–eGFP overexpression) MT growth in WT and KO NPCs using the EB3-tdTomato reporter that tracks growing plus ends of MTs (Fig 3I and Video 2). We found that AP-2-deficient NPCs generally had more EB3 tracks (Fig 3J, probably because of their tubular, elongated morphology; see also Fig 1E); however, these anterograde EB3 tracks originated further away from the centrosome (Fig 3K–M). Similarly, the number of retrograde tracks originating further distal from the centrosome was also significantly increased in NPCs lacking AP-2µ (Figs 3N and S3A). In line with the data above (see Fig 3D and E), AP-2 was not required for bulk MTs growth in these cells, because neither overall velocity and/or displacement of EB3 tracks was decreased upon AP-2 deletion (Fig S3B–D).

The centrosome and centrosome-anchored MTs are crucial for the radial migration of neocortical neurons (Kuijpers & Hoogenraad, 2011; Vinopal et al, 2023), and cells with non-centrosomal microtubule organization are typically polarized and nonmigratory (Bartolini & Gundersen, 2006). The polarized morphology of AP-2 KO NPCs (see Fig 1), combined with impaired centrosomal MT growth described above raised the question of whether AP-2 is required in NPCs for their migratory ability. Thus, we analyzed NPC migration using the neurosphere migration assay, which allows the assessment of radial migration of neurospheres after induction of neuronal differentiation (Fig S3E). We observed that the migration from neuroblasts was decreased in NPCs lacking AP-2 (Fig S3F and G). Taken together, these data suggest that AP-2 regulates MT nucleation at the centrosome and that this function is important for the migration of NPCs.

---

nuclei. Representative fluorescent images from N = 3. Scale bars: 6 µm. **(E)** Bar diagrams representing the colocalization of AP-2α and AP-2µ with GCP3 in NPCs based on Pearson's coefficient (r) (AP-2α = 0.78 ± 0.02, AP-2µ = 0.71 ± 0.037). In total, AP-2α = 20 AP-2µ = 19 cells from N = 3. A representative example of AP-2µ with GCP3 colocalisation is shown in Fig S2D. **(F)** WT and KO NPCs at DIV3 immunostained for GCP3 and co-immunostained for γ-tubulin. DAPI stain was used to visualize the nuclei. Representative fluorescent images from $N_{WT}$ = 3, $N_{KO}$ = 3. Scale bars: 20 µm, inserts 2 µm. **(G)** Levels of GCP3 at the centrosome, marked by γ-tubulin (mean grey value, m.g.v) in WT and KO NPCs (WT = 49,669 ± 1651, KO = 41,286 ± 2484, $P$ = 0.0086, two-tailed unpaired $t$ test). In total, 31 WT and 30 KO images from N = 3. **(H)** Percentage of total GCP3 overlapping with γ-tubulin in WT and KO NPCs (WT = 28.32 ± 2.62, KO = 11.53 ± 1.87, $P$ < 0.0001, two-tailed unpaired $t$ test). Total GCP3 was set to 100% in each condition. In total, 16 WT/KO images from N = 3. **(I)** WT and KO NPCs at DIV3 immunostained for PCNT. DAPI stain was used to visualize the nuclei. Representative fluorescent images from $N_{WT}$ = 3, $N_{KO}$ = 3. Scale bars: 10 µm. **(J)** Percentage of WT and KO NPCs with disorganized PCNT (WT = 0.83% ± 0.83%, KO = 66.52% ± 10.44%, $P$ = 0.0008, two-tailed unpaired $t$ test). In total 30 WT/KO cells, from N = 4. **(K)** WT and KO NPCs at DIV3 immunostained for PCM1 (arrows indicate PCM1 localization in WT and KO). DAPI stain was used to visualize the nuclei. Representative fluorescent images from $N_{WT}$ = 3, $N_{KO}$ = 3. Scale bars: 13 µm. **(L)** Analysis of PCM1 mean fluorescence (mean grey value, m.g.v) in WT and KO NPCs (WT = 36.03 ± 1.74, KO = 22.31 ± 1.13, $P$ < 0.0001, two-tailed unpaired $t$ test). In total, 30 WT/KO cells from N = 3. **(M)** Morphology of centrosomes in WT and KO NPCs visualized by immunostaining with γ-tubulin (arrows indicate KO NPCs containing multiple centrosomes). DAPI stain was used to visualize the nuclei. Scale bars: 10 µm. **(N)** Percentages of WT and KO NPCs which are either acentrosomal or containing single or multiple centrosomes at DIV3 (WT$^{Acentrosomal}$ = 10.98% ± 1.45%, KO$^{Acentrosomal}$ = 56.63% ± 4.61%, WT$^{SingleCentrosome}$ = 86.16% ± 1.58%, KO$^{SingleCentrosome}$ = 23.54% ± 2.42%, WT$^{MultipleCentrosomes}$ = 2.85% ± 0.74%, KO$^{MultipleCentrosomes}$19.82% ± 2.94%, $P$ < 0.0001, two-way ANOVA with Holm-Sidak's multiple comparisons test). In total, 15 WT/KO images, with >100 cells per condition, from N = 3. **(O)** Analysis of centrosomes in WT and KO NPCs, expressing either mCherry or AP-2µ-RFP and co-immunostained for CEP135 (please see Fig S2L for example images). Percentages of NPCs which are either acentrosomal or containing single or multiple centrosomes at DIV3 were counted (WT + mCherry$^{Acentrosomal}$ = 5.77% ± 2.38%, KO + mCherry$^{Acentrosomal}$ = 31.19% ± 4.10%, WT + AP-2µ-RFP$^{Acentrosomal}$ = 16.30% ± 3.88%, KO + AP-2µ-RFP$^{Acentrosomal}$ = 7.15% ± 3.59%, WT + mCherry$^{SingleCentrosome}$ = 94.23% ± 2.38%, KO + mCherry$^{SingleCentrosome}$ = 44.44% ± 6.74%, WT + AP-2µ-RFP$^{SingleCentrosome}$ = 81.33% ± 5.74%, KO + AP-2µ-RFP$^{SingleCentrosome}$ = 88.67% ± 1.36%, WT + mCherry$^{MultipleCentrosomes}$ = 0.00% ± 0.00%, KO + mCherry$^{MultipleCentrosomes}$24.37% ± 7.02%, WT + AP-2µ-RFP$^{MultipleCentrosomes}$ = 2.38% ± 2.38%, KO + AP-2µ-RFP$^{MultipleCentrosomes}$4.18% ± 2.65%). (WT + mCherry$^{Acentrosomal}$ versus KO + mCherry$^{Acentrosomal}$ $P$ = 0.001, WT + mCherry$^{Multiple}$ versus KO + mCherry$^{Multiple}$p = 0.002, WT + mCherry$^{Single}$ versus KO + mCherry$^{Single}$p<0.0001, KO + mCherry$^{Acentrosomal}$ versus KO + AP-2µ-RFP$^{Acentrosomal}$ $P$ = 0.002, KO + mCherry$^{Multiple}$ versus KO + AP-2µ-RFP$^{Multiple}$p = 0.007, KO + mCherry$^{Single}$ versus KO + AP-2µ-RFP$^{Single}$ $P$ < 0.0001, two-way ANOVA with Holm-Sidak's multiple comparisons test). >30 WT/KO images, from N = 3. **(P)** Morphology of centrosomes in NPCs either treated for 24 h with DMSO or with 30 µM of Pitstop2 visualized at DIV3 by immunostaining with γ-tubulin. DAPI stain was used to visualize the nuclei. Scale bars: 12 µm. **(Q)** Percentages of DMSO- and Pitstop2-treated NPCs which are either acentrosomal or containing a single centrosome or multiple centrosomes at DIV3 (DMSO$^{Acentrosomal}$ = 10.68% ± 3.35%, Pitstop2$^{Acentrosomal}$ = 6.51% ± 2.25%, DMSO$^{SingleCentrosome}$ = 87.54% ± 3.08%, Pitstop2$^{SingleCentrosome}$ = 92.14% ± 1.93%, DMSO$^{MultipleCentrosomes}$ = 1.78% ± 0.92%, Pitstop2$^{MultipleCentrosomes}$ = 1.35% ± 0.45%, two-way ANOVA with Holm–Sidak's multiple comparisons test). In total, 5 DMSO/Pitstop2 images, with ≥100 cells per condition, from N = 3. Data information: squares in Figs 2D, F, I, and M indicate magnified regions. All graphs show mean ± SEM. n.s.—nonsignificant. * indicates $P$ ≤ 0.05; ** indicates $P$ ≤ 0.01; *** indicates $P$ ≤ 0.001; **** indicates $P$ ≤ 0.0001. N represents independent cultures obtained from independent animals.

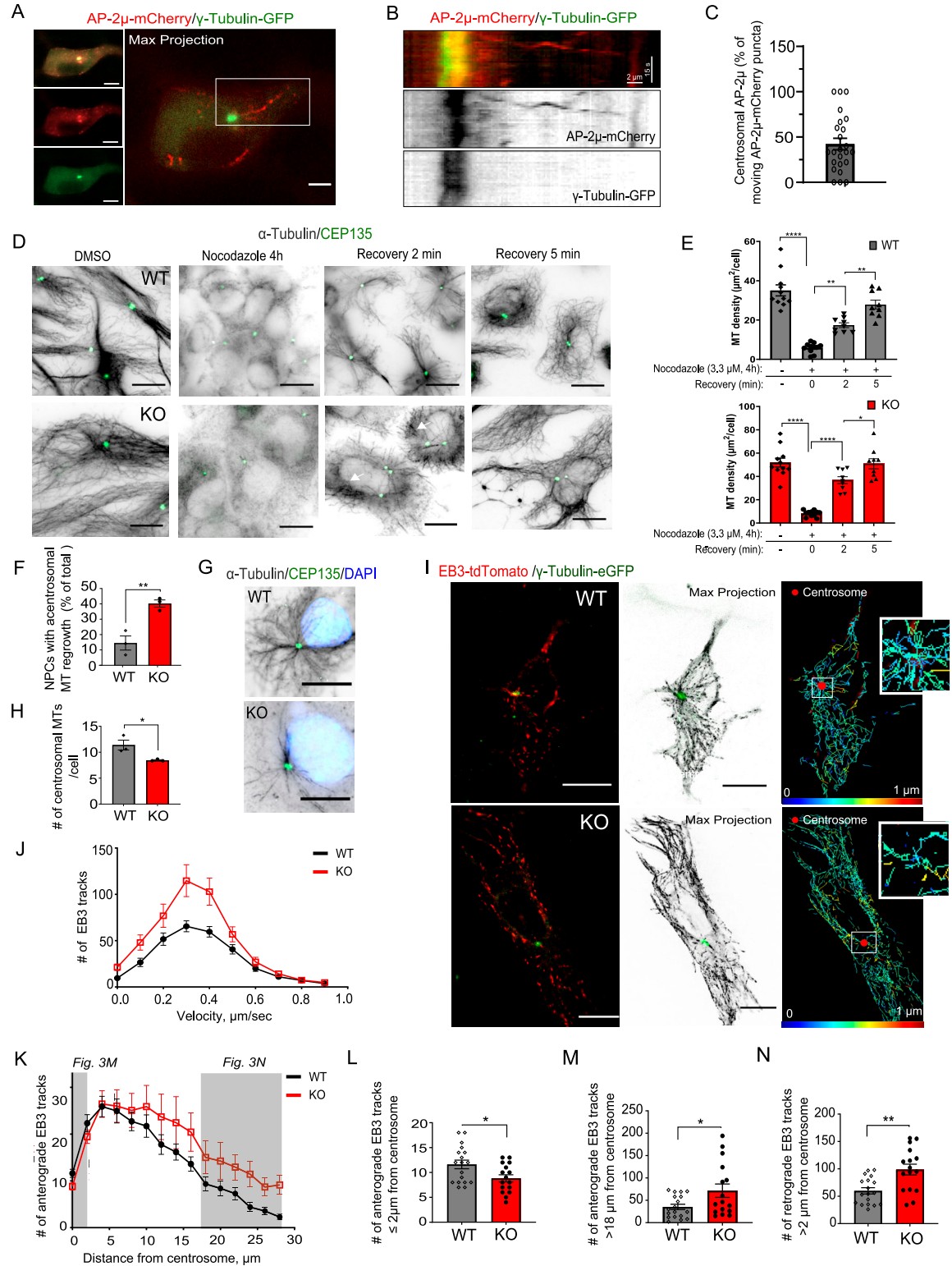

**Figure 3. AP-2 regulates centrosome function at the level of MT nucleation.**
**(A, B)** Representative time series (A) and corresponding kymographs (B) of AP-2μ-mCherry trafficking in NPCs at DIV3 co-transfected with γ-tubulin–GFP. Scale bars, (A) 3 μm, (B) 2 μm × 15 s. **(C)** Percentage of all AP-2μ-mCherry puncta trafficked towards the centrosome (42.21% ± 6.37%). In total, 24 NPCs. **(D)** Analysis of MT regrowth after their depolymerization with nocodazole (3.3 μM for 4 h) in WT and KO NPCs at DIV3. Representative fluorescent images indicate NPCs before the nocodazole treatment (control), immediately after the nocodazole washout (0 min) and after 2 and 5 min of nocodazole washout (2 min or 5 min). To visualize the MTs, NPCs were immunostained for α-tubulin; the centrosome was visualized immunostaining with CEP135. Scale bars: 20 μm. Arrows in "recovery 2 min" condition indicate sites of

## AP-2 depletion leads to spindle assembly defects, aberrant mitosis, and DNA damage

Centrosomal MTs play a key role in spindle orientation during mitosis (Lu & Johnston, 2013; di Pietro et al, 2016). Therefore, we next investigated whether loss of centrosomal MTs and disruption of centrosomes in AP-2 KO NPCs described above translates into defects in bipolar spindle assembly. Whereas most of the dividing WT NPCs formed a bipolar spindle, most AP-2-deficient NPCs in mitosis, but not Pistop2-treated NPCs, showed defects in spindle assembly, including the aberrant formation of multipolar and monopolar spindles (Fig 4A–D). To understand if spindle orientation defects result in aberrant mitoses, we examined mitotic figures in WT and KO NPCs using a phospho-histone H3 antibody as a chromosome condensation marker (Fig 4E and F). We observed a significant twofold increase in the total number of mitotic NPCs in AP-2 KO cells (Fig 4G), which were mostly in prometaphase, indicating a defect in mitotic progression (Fig 4H). However, the total number of G2/M cells was lower in KO NPCs, likely because of G2 cells exiting the cell cycle as a result of apoptosis (see Figs 1M and S4A and B). The inability to efficiently form bipolar mitotic spindles was also evident in a significant number of lagging chromosomes during anaphase (Fig 4I and J), a hallmark of chromosome segregation errors in mitosis. The effect of AP-2 loss on mitosis was independent of clathrin function in CME, because Pitstop2 had no effect on the number of mitotic NPCs (Fig S4C).

These data suggest that the loss of AP-2 compromises centrosomes and mitosis of NPCs leading to cell death and/or differentiation. In agreement with this, AP-2 KO NPCs accumulated DNA damage and were characterized by the stabilization of the stress-responsive transcription factor p53 (Fig 4K–N). Once p53 is stabilized, it is responsible for transcriptional activation of a number of proteins involved in apoptosis and/or senescence (Levine & Oren, 2009). Consistently, the deletion of AP-2 strongly increased the proportion of senescent NPCs (50% of total cell number), detected using colorimetric senescence-associated $\beta$-galactosidase (SA-$\beta$-gal) assay (Fig S4D and E).

Our data above indicate that most of the AP-2 KO NPCs withdraw from the cell cycle (likely at the G2/M transition, see Figs 1L and M

and S4A and B), whereas the remaining cells reveal defects in mitotic progression (see Fig 4G). Neuronal progenitors with prolonged mitosis are known to produce more neurogenic progeny (Pilaz et al, 2016). Thus, we next explored if the inhibition of cell cycle progression in AP-2 KO NPCs promotes their terminal differentiation by analysing the percentage of doublecortin-positive (Dcx) neurons, a marker of newborn migrating neurons, in WT and KO NPCs. We found that the percentage of KO NPCs committed to neurons at DIV3 and DIV5 was increased twofold compared with WT (Fig 4O–Q), suggesting that AP-2 is required to promote neuronal progenitor proliferation and inhibit neuronal differentiation. These data are also in agreement with drastic morphological alterations in KO NPCs from the WT round shape to an elongated tubular shape, which is a characteristic of terminally differentiated cells (see Fig 1).

## Proliferation defects in AP-2-deficient primary dividing cells can be rescued by inactivating p53

The data described thus far suggest that AP-2 is required for proliferation of neuronal progenitors by regulating mitotic progression and fidelity, and that its loss severely compromises their proliferation (see Fig 1). Because primary embryonic NPCs are difficult to maintain in culture and to test a different cell type, we next explored whether this AP-2 function could be reproduced in primary MEFs, which can be passaged and propagated longer in culture compared with NPCs. We isolated MEFs from E12.5 *Ap2m1* loxP: CAG-Cre[Tmx] mice with an identical genotype to the NPCs described above. Primary AP-2 KO MEFs (pMEFs) treated with Tmx for 3 d had significantly decreased AP-2 levels (Fig S5A and B), were impaired in their proliferation (Fig 5A), and were significantly delayed during prometaphase of mitosis (Fig 5B–D), although these defects were less pronounced when compared with primary AP-2 KO NPCs. Moreover, pMEFs were also characterized by defects in the bipolar spindle (Fig 5E and F) and showed lagging chromosomes during anaphase (Fig 5G and H). Consistent with the defects in centrosome morphology observed in primary NPCs, a significantly higher proportion of AP-2 KO pMEFs contained multiple

acentrosomal MT regrowth. **(E)** Analysis of MT density using $\alpha$-tubulin immunofluorescence in WT and KO NPCs before and after their depolymerization with nocodazole (WT$^{DMSO}$ = 35.09 ± 2.81, KO$^{DMSO}$ = 51.93 ± 3.81, WT$^0$ = 5.88 ± 0.76, KO$^0$ = 8.20 ± 0.81, WT$^2$ = 17.39 ± 1.23, KO$^2$ = 36.78 ± 3.00, WT$^5$ = 27.86 ± 2.24, KO$^5$ = 50.82 ± 4.32, pWT$^0$ versus WT$^{control}$ <0.0001, pWT$^0$ versus WT$^2$ = 0.0012, pWT$^2$ versus WT$^5$ = 0.006, pKO$^0$ versus KO$^{control}$ <0.0001, pKO$^0$ versus KO$^2$ < 0.0001, pKO$^2$ versus KO$^5$ = 0.027, one-way ANOVA with Tukey's multiple comparisons test). >9 images per condition, with each image containing >40 cells, from N = 3. **(F)** Percentages of WT and KO NPCs with acentrosomal MT regrowth after recovery from nocodazole treatment for 2 min (WT = 14.50 ± 4.59, KO = 40.20 ± 2.36, P = 0.008, two-tailed unpaired t test). In total, 14 WT/KO images, with each image containing >40 cells, from N = 3. **(G, H)** Representative images and analysis of centrosomal MT number in WT and KO NPCs, after recovery from nocodazole treatment for 2 min (WT = 11.38 ± 0.95, KO = 8.44 ± 0.11, P = 0.037, two-tailed unpaired t test). In total, 45 WT/KO cells from N = 3. Scale bar, 10 $\mu$m. **(I)** Representative time series (left panel) and maximum intensity projection images (middle panel) of 20 s (1 frame/1 s) of EB3-tdTomato- and $\gamma$-tubulin–GFP -expressing WT and AP-2 KO NPCs. Right panels indicate growing EB3 tracks from the images shown on the left using TrackMate/ImageJ plugin. EB3 track growth speed is color-coded. Red dots in the right panel indicate centrosome. Scale bars, 10 $\mu$m. **(J)** Histograms of EB3 tracks velocity ($\mu$m/s) in WT and KO NPCs. In total, 19 WT and 16 KO images per condition from N = 3. Please see mean velocity and total displacement of EB3 tracks in Fig S3B–D. **(K)** Histograms of the number of anterograde EB3 tracks plotted against the distance from the centrosome. Gray boxes indicate the distances quantified in Fig 3L and M. In total, 19 WT and 16 KO images per condition from N = 3. **(L)** Number of anterograde EB3 tracks originating in WT and AP-2 KO NPCs within ≤2 $\mu$m from the centrosome (WT = 11.61% ± 0.87%, KO = 8.81% ± 0.72%, P = 0.020, two-tailed unpaired t test). In total, 19 WT and 16 KO images per condition from N = 3. **(M)** Number of anterograde EB3 tracks originating in WT and AP-2 KO NPCs within >18 $\mu$m from the centrosome (WT = 35.22% ± 5.92%, KO = 71.44% ± 14.91%, P = 0.025, two-tailed unpaired t test). In total, 19 WT and 16 KO images per condition from N = 3. The distance of >18 $\mu$m likely corresponds to the elongated tubular morphology of KO NPCs (see Fig 1E). **(N)** Number of retrograde EB3 tracks originating in WT and AP-2 KO NPCs within >2 $\mu$m from the centrosome (WT = 59.83% ± 5.73%, KO = 98.63% ± 9.87%, P = 0.001, two-tailed unpaired t test). In total, 19 WT and 16 KO images per condition from N = 3. Please also see histograms of the number of retrograde EB3 tracks in Fig S3A. Data information: a square in Fig 3A indicates the region magnified in Fig 3B. Squares in Fig 3I indicate centrosomal EB3 tracks magnified to the right. All graphs show mean ± SEM. n.s.—nonsignificant. * indicates P ≤ 0.05; ** indicates P ≤ 0.01; *** indicates P ≤ 0.001; **** indicates P ≤ 0.0001.

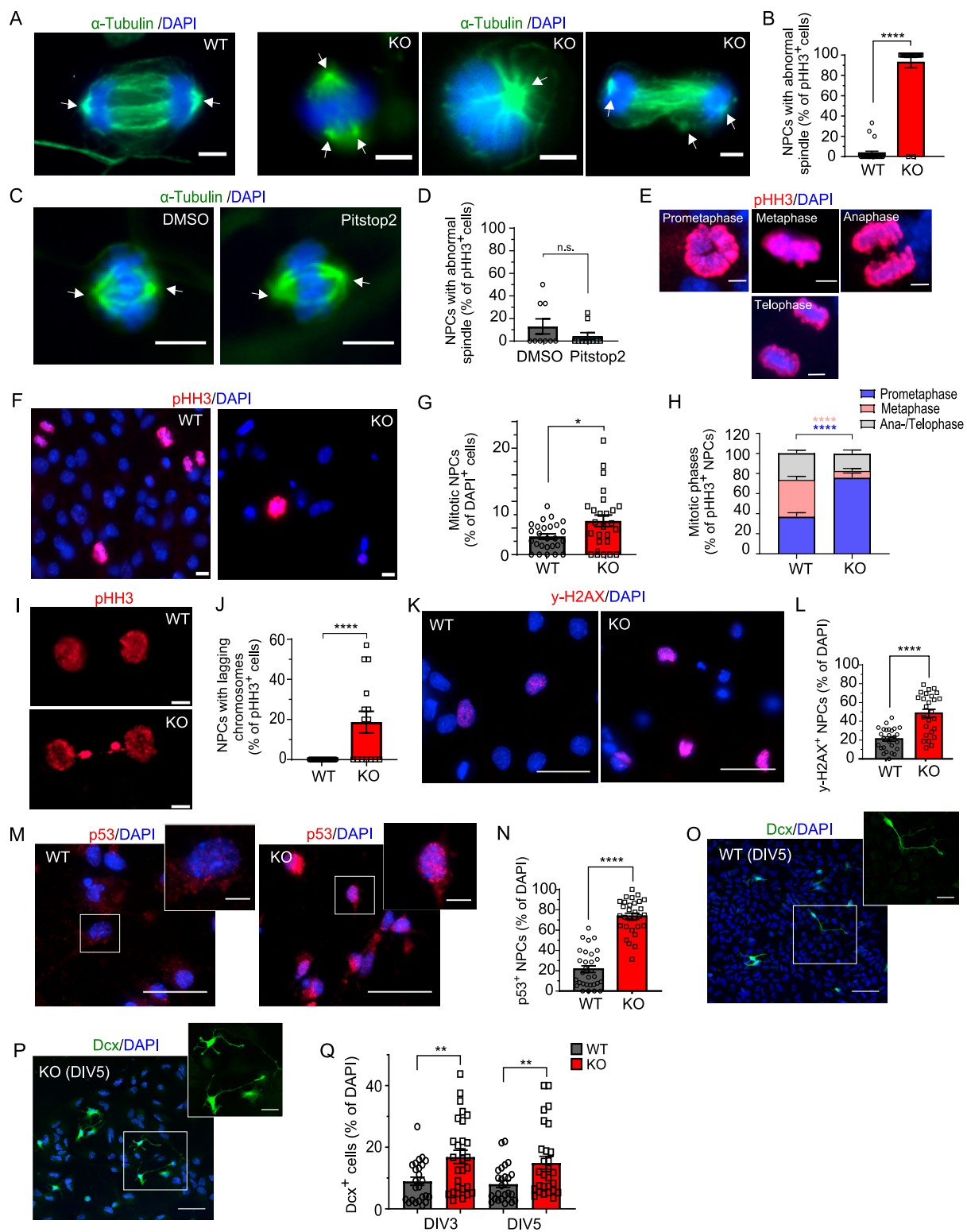

**Figure 4. AP-2 KO NPCs reveal spindle assembly defects and prolonged mitosis, but also show DNA damage and p53 activation.**
**(A)** Spindle morphology of WT and AP-2 KO NPCs revealed by immunostaining with α-tubulin. DAPI stain was used to visualize the nuclei. Scale bars: 3 μm. **(B)** Percentage of total pHH3⁺ WT and KO NPCs with abnormal spindle morphology (WT = 3.41% ± 1.92%, KO = 92.59% ± 5.14%, *P* < 0.0001, two-tailed unpaired *t* test). In total, 23 WT and 27 KO images, with at least 10 mitotic cells counted per condition, from N = 3. **(C)** Spindle morphology of WT NPCs treated for 24 h with 30 μM of Pitstop2 or DMSO as a control. Spindles were visualized using α-tubulin antibody. DAPI stain was used to visualize the nuclei. Scale bars: 5 μm. **(D)** Percentage of pHH3⁺ WT NPCs with abnormal spindles in DMSO- and Pitstop2-treated condition (Control = 12.96% ± 6.68%, Pitstop2 = 4.50% ± 3.02%, *P* = 0.248, two-tailed unpaired *t* test). In total, 9 WT and 10 KO images with at least 20 mitotic cells counted per condition. **(E)** Representative examples of mitotic phases in WT NPCs, visualized by immunostaining with pHH3. DAPI stain was

centrosomes (marked by both CEP135 and γ-tubulin immuno-staining) and/or were acentrosomal, although the latter phenotype was less pronounced compared with primary NPCs (Fig 5I and J). pMEFs lacking AP-2 were also impaired in their ability to progress through the cell cycle (Figs 5K and S5C–E) and revealed a decrease and disorganisation of centrosomal GCP2 and GCP3 localization (Figs 5L–O and S5F and G). Of note, in contrast to NPCs, we did not detect any dead cell fraction in the population of AP-2 KO pMEFs (Fig S5C, see also Fig 1L). These data and the fact that significantly more AP-2 KO pMEFs are in the G2/M phase of the cell cycle suggest that, unlike primary NPCs, these cells may not exit the cell cycle by apoptosis. We also observed that AP-2-deficient pMEFs grew significantly less centrosomal MTs after their release from nocodazole (Fig 5P–R) suggesting that the role of AP-2 in cell cycle progression and centrosome integrity is not NPC-specific.

Using pMEFs, we next investigated whether AP-2 loss compromises cell proliferation because of the inability of cells to progress through the cell cycle in a tumor suppressor-dependent manner. We speculated that immortalizing pMEF with the SV40 large T antigen, which binds and inactivates both retinoblastoma (Rb) and p53, might rescue the cell proliferation defects upon AP-2 loss (Fig 6A). In agreement with this hypothesis, immortalization of AP-2 KO pMEFs resulted in complete rescue of their mitotic index (Fig 6B and C) and improved the mitotic progression phenotype (Fig 6D). Immortalization of AP-2 KO pMEFs also abolished cell proliferation defects (Figs 6E and S6A–D). Interestingly, the rescue of these phenotypes in AP-2 KO iMEFs was independent of their ability to form bipolar spindle (Fig 6F and G), maintain centrosome integrity (Figs 6H and I and S6E and F) and properly segregate chromosomes during mitosis (Fig S6G) because these defects were still present in immortalized MEFs lacking AP-2. Finally, a decrease and disorganisation of centrosomal GCP2 and GCP3 localization (Fig S6H–M) and an impairment of centrosomal MT growth after nocodazole treatment (Fig 6J–L) were also evident in AP-2 KO iMEFs. Taken together, these data suggest that the cell cycle and mitotic defects in primary embryonic dividing cells lacking AP-2 can be rescued by driving them to bypass the Rb- and p53-dependent checkpoints, thus promoting cell cycle progression.

## Discussion

The role of the clathrin adaptor AP-2 in endocytosis and recycling of proteins at the plasma membrane has been intensively studied (Robinson, 2004; Kononenko & Haucke, 2015). Here, we provide evidence that AP-2 can also function at the centrosome. We find that the AP-2 complex associates with components of the γ-TuRC complex and regulates centrosomal MT nucleation. The function of AP-2 in regulating centrosome integrity is particularly important in primary NPCs, which rely on AP-2 to maintain their proliferative capacity and prevent premature differentiation. In addition, we found that AP-2 is also required for cell proliferation of primary MEFs and that this requirement can be bypassed by their immortalization with the SV40 large T antigen virus, which inactivates both Rb and p53 tumor suppressors. Numerous studies have previously knocked down AP-2 (mostly using RNAi-mediated approaches) to investigate its effect on CME (Motley et al, 2003; Diril et al, 2006; Keyel et al, 2008; Lau & Chou, 2008; Boucrot et al, 2010). In these studies, no apparent increase in apoptosis and/or cell division defects were reported, although cells depleted of AP-2μ have been shown to display aberrant spindles (Boucrot & Kirchhausen, 2007). However, most of the studies were performed in cancer or immortalized cell lines with disrupted cell cycle checkpoints. For instance, the transcription factor p53 is frequently mutated in cell lines from various types of cancers, which allows them to bypass the G1/S checkpoint of the cell cycle and divide indefinitely. Our data on the rescue of many AP-2-dependent cell cycle defects in MEFs with inactivated Rb and p53 explain why this requirement of AP-2 for cell cycle progression may have been overlooked in the past. Moreover, our results provide an alternative or additional explanation for the early embryonic lethality (around day 3.5 post-coitus or shortly thereafter) of mice that either completely lack AP-2μ or carry a mutation in the AP-2σ subunit of the AP-2 complex, a phenotype that has been solely attributed to the canonical role of AP-2 in CME (Mitsunari et al, 2005; Gorvin et al, 2017).

Several components of the CME machinery have previously been shown to regulate cell cycle progression in nonneuronal cells, but the cellular mechanism of their action remained controversial. Whereas some studies indicate an arrest at the G2/M phase and fragmented centrosomes in cells with knockdowns of CHC or GAK

---

used to visualize the nuclei. Scale bar, 3 μm. **(F)** Representative examples of WT and KO NPCs at DIV3 immunostained for pHH3. DAPI stain was used to visualize the nuclei. Scale bars: 6 μm. **(G)** Percentage of mitotic WT and KO NPCs at DIV3 (WT = 3.40% ± 0.50%, KO = 6.36% ± 1.09%, P = 0.017, two-tailed unpaired t test). In total, 26 WT/KO images with at least 100 cells per condition from N = 3. Please also the relative number of KO NPCs in the G2/M phase in Fig S4B. **(H)** Mitotic figures in WT and KO NPCs at DIV3 (WT[Prometaphase] = 37.16% ± 3.75%, KO[Prometaphase] = 77.45% ± 4.40%, WT[Metaphase] = 36.54% ± 3.50%, KO[Metaphase] = 6.24% ± 2.27%, WT[Ana-/Telo-phase] = 26.75% ± 2.88%, KO[Ana/Telpohase] = 16.24% ± 3.40%, pWT[Prometaphase] versus pKO[Prometaphase]<0.0001, pWT[Metaphase] versus pKO[Metaphase]<0.0001, two-way ANOVA with Holm–Sidak's multiple comparisons test). In total, 37 WT and 41 KO images with in total >80 mitotic cells pro condition from N = 3. **(I)** Representative examples of lagging chromosomes in KO NPCs, visualized using pHH3 antibody. Scale bars: 3 μm. **(J)** Percentage of pHH3+ WT and KO NPCs with lagging chromosomes at DIV3 (WT = 0.00% ± 0.00%, KO = 18.67% ± 5.42%, P < 0.0001, two-tailed unpaired t test). In total, 29 WT and 15 KO images which contained cells in Ana-/Telophase, with total >80 mitotic cells pro condition from N = 3. **(K)** Representative images of WT and AP-2 KO NPCs immunostained for γ-H2AX. DAPI stain was used to visualize the nuclei. Scale bars: 30 μm. **(L)** Percentage of γ-H2AX+ WT and AP-2 KO NPCs (WT = 21.14% ± 2.31%, KO = 48.54% ± 4.24%, P < 0.0001, two-tailed unpaired t test). In total, 26 WT/KO images, with at least 100 cells counted per condition from N = 3. **(M)** Representative images of WT and AP-2 KO NPCs immunostained for p53. DAPI stain was used to visualize the nuclei. Scale bars: 30 μm (5 μm inserts). **(N)** Percentage of p53+ WT and AP-2 KO NPCs (WT = 21.53% ± 3.33%, KO = 73.60% ± 3.05%, P < 0.0001, two-tailed unpaired t test). In total 30 WT/KO, with at least 100 cells counted per condition from N = 3. **(O, P)** Representative images of WT and KO NPCs at DIV5, grown in the media, which induces spontaneous differentiation and immunostained for doublecortin X (Dcx). DAPI stain was used to visualize the nuclei. Scale bars: 50 μm (20 μm inserts). **(Q)** Percentage of Dcx-positive cells among WT and KO NPCs at DIV3 and DIV5 (WT[DIV3] = 8.97% ± 1.34%, KO[DIV3] = 16.87% ± 2.15%, WT[DIV5] = 8.07% ± 1.13%, KO[DIV5] = 14.95% ± 2.10%, pWT[DIV3] versus pKO[DIV3] = 0.005, pWT[DIV3] versus pKO[DIV3] = 0.007, two-tailed unpaired t test). ≥24 WT/KO images with >70 total cells counted per condition, from N = 3. Data information: squares in Fig 4M and O indicate the regions magnified. Arrows in Fig 4A and 4C indicate mitotic spindles. All graphs show mean ± SEM. n.s.—nonsignificant; * indicates P ≤ 0.05; ** indicates P ≤ 0.01; *** indicates P ≤ 0.001; **** indicates P ≤ 0.0001. N represents independent cultures obtained from independent animals.

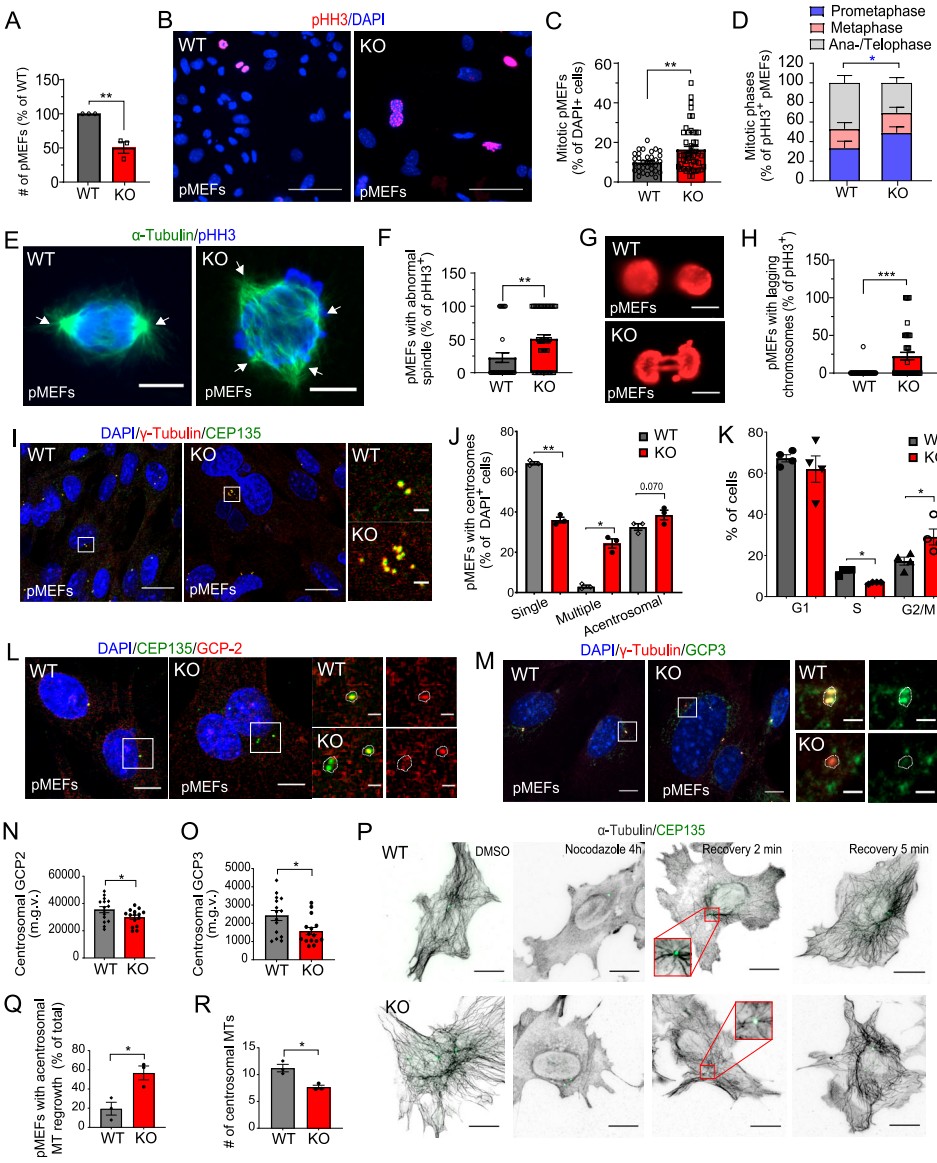

**Figure 5. Proliferation defects and loss of centrosomal integrity are observed in AP-2μ-deficient primary MEFs.**
**(A)** Relative number of WT and AP-2 KO primary MEFs at DIV3 (WT set to 100%, KO = 50.853% ± 8.40%, $P$ = 0.0042, $N_{WT}$ = 3, $N_{KO}$ = 3, two-tailed unpaired $t$ test). **(B)** Representative examples of WT and KO primary MEFs at DIV3 immunostained for pHH3. DAPI stain was used to visualize the nuclei. Scale bars: 63 μm. **(C)** Percentage of mitotic WT and KO primary MEFs (WT = 9.97% ± 0.82%, KO = 16.45% ± 1.62%, $P$ = 0.002, two-tailed unpaired $t$ test). In total, 33 WT and 49 KO images with at least 100 cells per condition from N = 3. **(D)** Mitotic figures in WT and KO primary MEFs (WT$^{Prometaphase}$ = 30.93% ± 5.85%, KO$^{Prometaphase}$ = 52.87% ± 5.68%, WT$^{Metaphase}$ = 19.44% ± 5.02%, KO$^{Metaphase}$ = 15.82% ± 4.57%, WT$^{Ana-/Telo-phase}$ = 49.63% ± 6.12%, KO$^{Ana/Telophase}$ = 31.32% ± 5.12%, pWT$^{Prometaphase}$ versus pKO$^{Prometaphase}$ = 0.046, two-way ANOVA with Holm–Sidak's multiple comparisons test). In total, 45 WT and 59 KO images from N = 4. **(E)** Spindle morphology of WT and AP-2 KO primary MEFs revealed by immunostaining with α-tubulin and pHH3 antibodies. Scale bars: 6.5 μm. **(F)** Percentage of pHH3+ WT and KO primary MEFs with abnormal spindle morphology (WT = 22.73% ± 7.247%, KO = 51.04% ± 6.11%, $P$ = 0.004, two-tailed unpaired $t$ test). In total 33 WT and 49 KO images per condition from N = 3. **(G)** Representative examples of lagging chromosomes in DIV3 KO primary MEFs, visualized by pHH3 antibody. Scale bars: 6.5 μm. **(H)** Percentage of pHH3+ WT and KO primary MEFs with lagging chromosomes (WT = 0.97% ± 0.97%, KO = 21.83% ± 4.82%, $P$ = 0.0004, two-tailed unpaired $t$ test). In total 32 WT and 31 KO images per condition from N = 3. **(I)** Morphology of the centrosomes in WT and KO pMEFs visualized by co-immunostaining with γ-tubulin and CEP135. DAPI stain was used to visualize the nuclei. Scale bars: 20 μm. **(J)** Percentages of WT and KO pMEFs which are either acentrosomal or containing single or multiple centrosomes at DIV3 (WT$^{Acentrosomal}$ = 32.54% ± 1.62%, KO$^{Acentrosomal}$ = 38.53% ± 2.38%, WT$^{SingleCentrosome}$ = 64.47% ± 0.99%, KO$^{SingleCentrosome}$ = 37.11% ± 1.31%,

WT$^{MultipleCentrosomes}$ = 2.99% ± 0.70%, KO$^{MultipleCentrosomes}$24.36% ± 2.28%, pWT$^{SingleCentrosome}$ versus KO$^{SingleCentrosome}$ = 0.007, pWT$^{Multiple}$ versus KO$^{Multiple}$ = 0.03, two-way ANOVA with Holm–Sidak's multiple comparisons test). In total, 60 WT/KO images, with >100 cells per condition, from N = 3. **(K)** Cell cycle analysis by DNA content estimation with flow cytometry in WT and AP-2 KO pMEFs. The graph indicates the percentages of cells in G1, S, and G2/M phases of the cell cycle (WT$^{G1}$ = 67.25% ± 1.75%, KO$^{G1}$ = 62.00% ± 6.42%, $P$ = 0.350, WT$^{S}$ = 12.25% ± 0.75%, KO$^{S}$ = 6.75% ± 0.25%, $P$ = 0.023, WT$^{G2/M}$ = 17.25% ± 1.93%, KO$^{G2/M}$ = 29.00% ± 3.87%, $P$ = 0.038, two-way ANOVA with Holm–Sidak's multiple comparisons test). In addition, significant differences were found between WT and KO in the overall distribution of cell cycle phases with the $x^2$ test using the original cell counts ($P < 0.001$ for each independent N). WT/KO = 20,000 events, from N = 4. Please see also representative histograms in Fig S5C. **(L)** WT and KO pMEFs at DIV3 immunostained for GCP2 and co-immunostained for CEP135. DAPI stain was used to visualize the nuclei. Representative fluorescent images from N = 3. Scale bars: 10 μm, inserts 2 μm. **(M)** WT and KO pMEFs at DIV3 immunostained for GCP3 and co-immunostained for γ-tubulin. DAPI stain was used to visualize the nuclei. Representative fluorescent images from N = 3. Scale bars: 10 μm, inserts 2 μm. **(N)** Levels of GCP2 at the centrosome, marked by CEP135 (mean grey value, m.g.v) in WT and KO pMEFs (WT = 49,669 ± 1651, KO = 41,286 ± 2484, $P$ = 0.0086, two-tailed unpaired $t$ test). In total 15 WT/KO images from N = 3. **(O)** Levels of GCP3 at the centrosome, marked by γ-tubulin (mean grey value, m.g.v) in WT and KO pMEFs (WT = 35,464 ± 2226, KO = 29,871 ± 1557, $P$ = 0.049, two-tailed unpaired $t$ test). In total, 15 WT/KO images from N = 3. **(P)** Analysis of MT regrowth after their depolymerization with nocodazole (3.3 μM for 4 h) in WT and KO pMEFs at DIV3. Representative fluorescent images indicate pMEFs before the nocodazole treatment (control), immediately after the nocodazole washout (0 min) and after 2 and 5 min of nocodazole washout (2 or 5 min). To visualize the MTs, pMEFs were immunostained for α-tubulin, the centrosome was visualized immunostaining with CEP135. Scale bars: 20 μm. **(Q)** Percentages of WT and KO pMEFs with acentrosomal MT regrowth after recovery from nocodazole treatment for 2 min (WT = 19.49 ± 6.67, KO = 56.73 ± 7.26, $P$ = 0.019, two-tailed unpaired $t$ test). In total, 41 WT/KO images, with each image containing >40 cells, from N = 3. **(R)** Analysis of centrosomal MT number in WT and KO pMEFs, after recovery from nocodazole treatment for 2 min (WT = 11.21 ± 0.70, KO = 7.67 ± 0.35, $P$ = 0.010, two-tailed unpaired $t$ test). In total, 34 WT/KO cells from N = 3. Scale bar, 10 μm. Data information: squares in Figs 5I, L, M, and P indicate regions magnified. Arrows in Fig 5E indicate mitotic spindles. All graphs show mean ± SEM. n.s.—nonsignificant; * indicates $P ≤ 0.05$; ** indicates $P ≤ 0.01$; *** indicates $P ≤ 0.001$; **** indicates $P ≤ 0.0001$. N represents independent cultures obtained from independent animals.

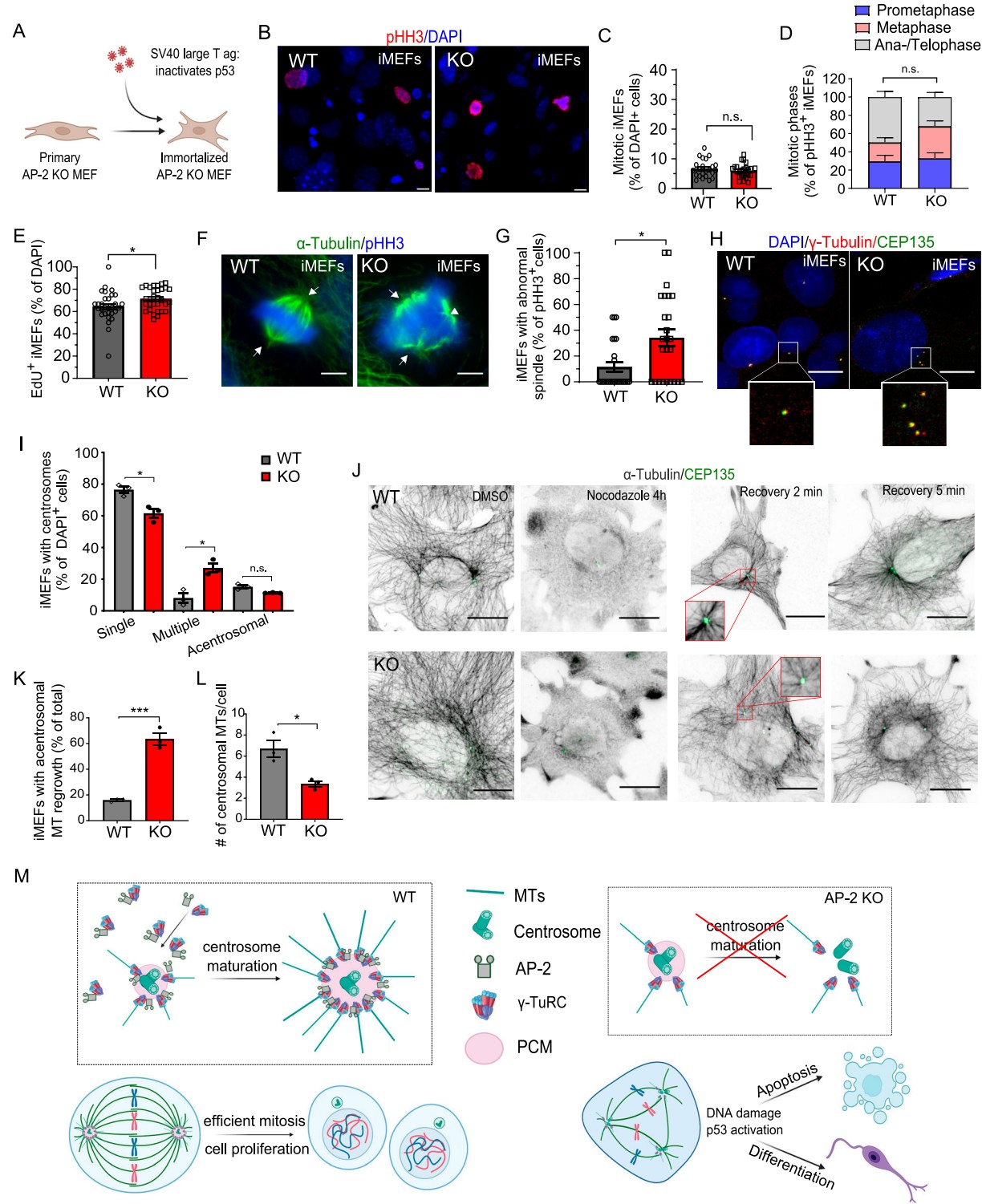

**Figure 6. Proliferation defects in AP-2μ-deficient MEFs are rescued by inactivating the Rb- and p53-dependent checkpoints.**
**(A)** Schematic illustration of immortalization of primary MEFs using the simian vacuolating (SV) 40 large T antigen virus. **(B)** Representative examples of WT and KO immortalized MEFs (iMEFs) immunostained for pHH3. DAPI stain was used to visualize the nuclei. Scale bars: 5 μm. **(C)** Percentage of mitotic WT and KO iMEFs (WT = 6.64% ± 0.60%, KO = 5.97% ± 0.7%, P = 0.400, two-tailed unpaired t test). In total 24 WT and 25 KO images with > 67 mitotic cells per condition from N = 3. **(D)** Mitotic figures in WT and KO iMEFs (WT$^{Prophase}$ = 29.80% ± 6.41%, KO$^{Prophase}$ = 33.13% ± 5.83%, WT$^{Metaphase}$ = 20.73% ± 4.93%, KO$^{Metaphase}$ = 35.07% ± 5.77%, WT$^{Ana-/Telo-phase}$ = 49.47% ± 6.23%, KO$^{Ana/Telopohase}$ = 31.80% ± 5.03%, pWT$^{Metaphase}$ versus pWT$^{Ana-/Telo-phase}$ = 0.008, two-way ANOVA with Holm–Sidak's multiple comparisons test). In total 25 WT/KO images from N = 3. **(E)** Percentage of proliferating WT and KO iMEFs (EdU⁺ cells) (WT = 64.51% ± 2.52%, KO = 71.29% ± 1.83%, P = 0.034, two-tailed unpaired t test). In total, 30 WT/KO images, with at least 300 EdU cell and 1500 total cells per condition, from N = 3. Please see the corresponding images in Fig S6D. **(F)** Spindle morphology of WT and AP-2 KO iMEFs

(an Hsc70-cochaperone required for uncoating of clathrin-coated vesicles) (Royle et al, 2005; Shimizu et al, 2009; Tanenbaum et al, 2010; Foraker et al, 2012), others show that the cells remain in G1 phase (Sigismund et al, 2008; Sousa et al, 2012). Here, we describe that in primary neuronal progenitors AP-2 deletion results in the loss of centrosome integrity and severely impacts mitotic progression and fidelity. The disruption of centrosome structure and/ or function is known to activate a "mitotic surveillance pathway" that leads to p53-dependent cell cycle arrest in G1 and/or cell death, which are largely independent of aneuploidy or DNA damage (Mikule et al, 2007; Bazzi & Anderson, 2014; Insolera et al, 2014; Lambrus et al, 2015; Wong et al, 2015). Therefore, we favor the hypothesis that loss of AP-2 compromises centrosome function, which in turn leads to cell cycle arrest (senescence and differentiation) and/or cell death, but also causes additional phenotypes such as chromosome segregation errors and DNA damage. Exactly how deletion of AP-2 in NPCs results in lagging chromosomes during anaphase and whether this is a consequence of the centrosome amplification and/or multipolar spindle formation observed in a proportion of AP-2-depleted NPCs are currently unknown. Possible links may be established by BubR1, a key component of the spindle assembly checkpoint, which has previously been shown to interact with the $\beta$ subunit of the AP-2 complex (Cayrol et al, 2002).

We observed a similar phenotype of growth arrest and centrosome integrity defects in AP-2 KO pMEF cells, akin to the GAK KO MEFs reported previously (Olszewski et al, 2014). Interestingly, GAK KO MEFs, and HeLa cells with CHC or AP-2α knockdowns do not show induction of apoptosis but rather become senescent (Hinrichsen et al, 2003; Olszewski et al, 2014). In our study, we found that AP-2-deficient NPCs are more prone to cell death compared with AP-2 KO pMEFs, but can also undergo senescence and/or differentiate. It is not clear what determines the choice between cell death and cell differentiation in these cells, but the mechanisms may be related in part to the p53 signaling axis, which can trigger either the cell elimination or cell survival program depending on cell type and persistence of DNA damage (Murray-Zmijewski et al, 2008; Kruiswijk et al, 2015). Interestingly, when activated in stem cells, p53 promotes their differentiation by arresting cell cycle progression (Levine et al, 2016). It is conceivable that the levels of p53 or dynamics of its stabilization in NPCs lacking AP-2 determine the heterogeneity of

the response to DNA damage and, depending on their cell division mode, either transactivates genes that stop proliferation and trigger differentiation or induces cell death (Purvis et al, 2012).

How does the AP-2 complex organize the centrosome? A possible function of AP-2 could be to mediate the intracellular trafficking of centrosome components. We have previously found that AP-2 associates with the p150Glued component of the dynactin complex and that this function is crucial for the transport of autophagosomes along MTs in mature neurons (Kononenko et al, 2017). Here, we extend our knowledge on the trafficking functions of AP-2 in embryonic neuronal cells by providing evidence of its association with components of γ-TuRC complex at the centrosome. Centrosomes are major sites for MT nucleation in cells, and assembly of the active γ-TuRC at the centrosome is a crucial event to obtain fully functional MT nucleating complexes required for efficient mitotic spindle assembly (Schiebel, 2000; Guichard et al, 2010; Teixidó-Travesa et al, 2010; Guillet et al, 2011). The assembly of the γ-TuRC is achieved by the association of a core structural subunit (γ-TuSC, γ-tubulin small complex) composed of γ-tubulin, GCP2, and GCP3 with additional proteins, which depending on the organism include GCP4-GCP6, NEDD1/GCP-WD, and MOZART2/GCP8 (Kollman et al, 2010; Paz & Lüders, 2018; Böhler et al, 2021). Here, we find that AP-2 associates with γ-TuSC components and GCP4 and GCP8 in embryonic and adult brains. This finding, together with our observation that the AP-2μ subunit traffics to the centrosome, suggests a model whereby AP-2 regulates centrosome function at the level of MT nucleation by associating with components of the γ-TuRC complex. We propose that impaired MT nucleation from the centrosome in AP-2 KO NPCs activates the spindle assembly checkpoint, prolongs mitosis, and causes cell cycle arrest and/or cell death, as it has been previously reported (Müller et al, 2006). Another possible interpretation of our data is that AP-2 in the cytoplasm acts as a preassembly station for the components of the γ-TuRC complex before their recruitment to the centrosome. The two interpretations of AP-2 function are not mutually exclusive. Several centrosome proteins are not restricted to the centrosome but are also present in the cytoplasm (Moudjou et al, 1996; Paoletti et al, 1996). For instance, in interphase cells, the preassembled γ-TuRC is located in the cytoplasm (Vérollet et al, 2006), and is recruited to the pericentriolar material of centrosomes upon mitotic entry (Haren et al, 2009). We speculate that AP-2 could

---

revealed by immunostaining with α-tubulin and pHH3 antibodies. Scale bars: 6.5 μm. **(G)** Percentage of pHH3+ WT and KO iMEFs with abnormal spindle morphology (WT = 11.47% ± 3.70%, KO = 34.27% ± 6.56%, $P$ = 0.004, two-tailed unpaired $t$ test). In total 25 WT/KO images per condition from N = 3. **(H)** Morphology of centrosomes in WT and KO iMEFs visualized by co-immunostaining with γ-tubulin and CEP135. DAPI stain was used to visualize the nuclei. Scale bars: 20 μm. **(I)** Percentages of WT and KO iMEFs which are either acentrosomal or containing single or multiple centrosomes at DIV3 (WT$^{Acentrosomal}$ = 15.19% ± 1.01%, KO$^{Acentrosomal}$ = 11.37% ± 0.16%, WT$^{SingleCentrosome}$ = 76.48% ± 2.12%, KO$^{SingleCentrosome}$ = 61.57% ± 2.94%, WT$^{MultipleCentrosomes}$ = 8.10% ± 2.87%, KO$^{MultipleCentrosomes}$27.06% ± 2.78%, pWT$^{SingleCentrosome}$ versus KO$^{SingleCentrosome}$ = 0.0360, pWT$^{Multiple}$ versus KO$^{Multiple}$ = 0.024, two-way ANOVA with Holm–Sidak's multiple comparisons test). In total >45 WT/KO images, with >50 cells per condition, from N = 3. **(J)** Analysis of MT regrowth after their depolymerization with nocodazole (3.3 μM for 4 h) in WT and KO iMEFs. Representative fluorescent images indicate iMEFs before the nocodazole treatment (control), immediately after the nocodazole washout (0 min) and after 2 and 5 min of nocodazole washout (2 or 5 min). To visualize the MTs, iMEFs were immunostained for α-tubulin; the centrosome was visualized immunostaining with CEP135. Scale bars: 20 μm. **(K)** Percentages of WT and KO iMEFs with acentrosomal MT regrowth after recovery from nocodazole treatment for 2 min (WT = 15.87 ± 0.79, KO = 63.41 ± 4.70, $P$ = 0.0006, two-tailed unpaired $t$ test). In total, 30 WT/KO images, with each image containing >100 cells, from N = 3. **(L)** Analysis of centrosomal MT number in WT and KO iMEFs, after recovery from nocodazole treatment for 2 min (WT = 6.70 ± 0.80, KO = 3.36 ± 0.28, $P$ = 0.010, two-tailed unpaired $t$ test). In total, 80 WT/KO cells from N = 3. Scale bar, 10 μm. **(M)** Hypothetical model of how AP-2 regulates cell proliferation of NPCs. In the WT condition, AP-2 associates with γ-TuRC components and localizes to the centrosomes, where it regulates the growth of centrosomal MTs, a function essential for bipolar spindle formation and mitotic progression. Loss of AP-2 impairs the localization of γ-TuRC to the centrosome, impairing its function at the level of MT nucleation, and leads to either premature neuronal differentiation or cell death. Data information: squares in Fig 6H and J indicate the regions magnified. Arrows in Fig 6F indicate mitotic spindles. All graphs show mean ± SEM. n.s.—nonsignificant; * indicates $P$ ≤ 0.05; ** indicates $P$ ≤ 0.01; *** indicates $P$ ≤ 0.001; **** indicates $P$ ≤ 0.0001. N represents independent cultures obtained from independent animals.

contribute to the expansion of pericentriolar material at the onset of mitosis by recruiting additional γ-TuRC complexes, thereby increasing MT nucleation activity as the cell cycle progresses; however, more data are required to support this hypothesis. Recently, it was shown that the γ-TuRC localization is not limited to the PCM but is found inside the centriole cylinder, where it additionally regulates centriole stabilization (Schweizer et al, 2021). It remains to be determined whether AP-2-dependent γ-TuRC trafficking controls centriole stabilization or integrity.

Taken together, to the best of our knowledge, we report the first demonstration of a cell cycle role for the endocytic adaptor AP-2 in proliferating neuronal cells. We identify AP-2 as a novel component of the centrosome, and propose that AP-2 is important both for the maintenance of centrosome integrity and centrosomal MT nucleation (Fig 6M). This function of AP-2 is independent of the CME, as the observed phenotypes were not present in NPCs treated with the clathrin inhibitor Pitsop2, which inhibits the association of clathrin with clathrin box motif-containing CME proteins (von Kleist et al, 2011) and/or in NPCs in which clathrin function was inhibited by shRNA-mediated knockdown. Whether clathrin functions at the centrosome and regulates mitotic progression of NPCs, similar to its previously described noncanonical role in nonneuronal cells (Royle et al, 2005), is currently unknown. It is equally unknown whether AP-2-dependent regulation of centrosomal MT nucleation is required in the early stages of embryonic brain development, a question to be resolved in the future. The role of AP-2 in centrosome integrity is likely not limited to mouse cells, as AP-2α has been previously identified as a regulator of centrosome overamplification in human cells (Balestra et al, 2013). These data, together with the fact that AP-2 expression has recently been linked to several types of cancer (Cho et al, 2019; Wu et al, 2020), open an exciting new perspective for therapeutic approaches to cancer treatment by targeting the function of AP-2 in cell proliferation. Equally intriguing is the investigation of the centrosome-organizing role of AP-2 in the context of Alzheimer's disease, a neurodegenerative condition, which is characterized by both decreased neurogenesis (Moreno-Jiménez et al, 2019), and aberrant cell cycle events (Arendt, 2012; Mertens et al, 2021). Because AP-2 levels are reduced in the Alzheimer's disease brain (Seyfried et al, 2017), we suggest that the components of the AP-2 complex should be explored with respect to their function in regulating cell cycle events in the mature brain.

# Materials and Methods

## Animals

C57/BL/6 mice were housed in polycarbonate cages at standard 12/12 d–night cycles, and water and food were provided ad libitum. All animal experiments were approved by the Ethics Committee of LANUV Cologne NRW and are conducted according to the committee's guidelines (AZ 81-02.04.2020.A418, AZ 84-02.04.2016.A041, AZ 81-02.05.40.20.075, AZ 84-02.04.2016.A451). Conditional tamoxifen-inducible AP-2μ KO mice (*Ap2m1*lox/lox:CAG-Cre[Tmx]) are described in Kononenko et al (2017) and were used for in-vitro experiments. Primers used for mouse genotyping are shown in Supplemental Data

1. C57BL6/NRj were used for WT experiments (MS, Co-IPs, colocalization, and trafficking experiments).

## Cell culture of primary NPCs

Before embryo preparation, coverslips and dishes were coated with laminin (10 μg/ml) either at 37°C for 2 h or overnight at 4°C. Embryos were isolated between E12–E14 of pregnancy. The telencephalic vesicles were extracted and digested for 20 min in 0.05% trypsin supplemented with DNAse. The addition of soybean trypsin inhibitor stopped trypsinization, and tissue was homogenized by pipetting gently 10 times with a p1000 and 10 times with a p200. Homogenates were centrifuged (200*g*, 1 min) and washed once before pellet resuspension in proliferation media. Cell concentration was adjusted at a concentration of 50,000 cells/ml for stock and 10,000 cells/ml for experiments and plated on laminin-coated coverslips/dishes. Once the cells were attached to the surface, 2 ml of pre-warmed proliferation media (serum-free DMEM/F-12 & GlutaMAX [Gibco]; 2% B27 [Gibco]; 20 ng/ml EGF; 10 ng/ml b-FGF, 1% penicillin/streptomycin) was added to each well/dish. Homologous recombination in *Ap2m1*lox/lox:CAG-Cre[Tmx] NPCs was induced by applying 0.4 μM (Z)-4-hydroxytamoxifen (Sigma-Aldrich) of tamoxifen (or equal concentration of ethanol) immediately after plating. Differentiation of NPCs was induced by differentiation media (MEM medium [Gibco], 30 mM D-Glucose, 2 mM NaHCO₃, 5% FBS, 0.25% GlutaMAX [Gibco], 1% penicillin/streptomycin, 2% B27 [Gibco], 100 mg/liter Transferrin). NPCs were maintained in an incubator at 37°C and 5% $CO_2$ until being used in experiments.

## NPCs plasmid transfection

NPCs were transfected using Lipofectamine LTX (Invitrogen) following the manufacturer's guidelines for mouse NPCs. In brief, 1.5 μg plasmid DNA (see Supplemental Data 2), 3 μl of Plus Reagent, and 4 μl of Lipofectamine LTX reagent were mixed in Opti-MEM medium (200 μl) (for each well of a six-well plate). It was followed by incubation for 10 min allowing for lipo-complex formation at room temperature. Lipo-complexes were added to the cells and incubated at 37°C, 5% $CO_2$ for 48 h, depending on the plasmid expression. For experiments where the cytosolic overexpression marker needed to be visualized together with MTs or cytoskeletal proteins, transfected cells were first fixed with first 4% PFA for 3 min and immediately after that with pre-cooled 100% methanol in −20°C for 5 min. Afterwards, the cells were washed with 1x PBS three times.

## Cell culture of primary MEFs

Embryos were obtained between stages E12–E14 of pregnancy. Head, visceral tissues, and gonads were removed from the embryos, and the carcass was digested in 0.25% trypsin/1 mM EDTA for 20 min at 37°C. Trypsinization was stopped by adding MEF culture medium (DMEM & GlutaMAX [Gibco], 10% FBS, 0.1% penicillin/streptomycin) and the tissue was homogenized by pipetting up and down. Samples were incubated for 5 min at RT before centrifugation (200*g*, 5 min). The pellet was resuspended in MEF culture medium, and the

cells were plated. The cells were incubated at 37°C with 5% $CO_2$ until they reached confluency.

## SV-40 immortalization of MEFs

SV40 virus (kindly provided by Dr. Frederik Tellkamp, Marcus Krüger lab, CECAD, Cologne) was added to the cells together with polybrene to enhance the survival and transduction efficiency. The culture medium was exchanged 12 h after transduction. Cells are considered immortalized after passage 10 and the presence of SV40 validated with PCR.

## Immunostaining

Cells were either fixed in 4% PFA dissolved in PBS (137 mM NaCl, 2.7 mM KCl, 10 mM $Na_2HPO_4$, 1.8 mM, $K_2HPO_4$, pH 7.4) for 15 min at RT or 8 min in ice-cold methanol at −20°C, depending on the protein of interest and processed as described previously (Kononenko et al 2017).

## Pitstop2-mediated CME inhibition

Treatment with Pitstop2 blocks ligand access to the clathrin terminal domain, hence inhibiting CME. 1 h after plating, 1 $\mu$l of 30 mM Pitstop2 (Abcam) or DMSO as control was added per 1 ml of serum-free cell culture media. 24 or 48 h after treatment cells were used for experiments. Pistopt2 at a concentration of 30 $\mu$M for 6 h was previously shown to induce a mitotic phenotype because of the inhibition of clathrin in HeLa cells (Smith et al, 2013).

## Transferrin uptake assay

For transferrin uptake, the media were removed and pre-warmed serum free DMEM/F-12, GlutaMAX was added to the wells. Cells were starved in these media for 1 h at 37°C. They were then incubated for 45 min at 37°C with a serum-free medium containing 15 $\mu$g/ml human Tf conjugated to Alexa Fluor 488 (Invitrogen). Cell surface-bound Tfn was removed by an ice-cold acid wash (0.2 M acetic acid + 0.5 M NaCl, pH 2.8) for 5 min then rinsed with PBS 2 times. Cells were immediately fixed with 4% PFA for 15 min at room temperature. For Pitstop2 treatment, cells were incubated with 30 $\mu$M pitstop2 for 24 or 48 h and afterwards were starved in pre-warmed serum-free DMEM/F-12, GlutaMAX for 1 h. Then, 15 $\mu$g/ml human Tf conjugated to Alexa Fluor 488 (Invitrogen) was added to the wells and incubated for 45 min. Cell surface-bound Tfn was removed by an ice-cold acid wash (0.2 M acetic acid + 0.5 M NaCl, pH 2.8) for 5 min then rinsed with PBS two times. Cells were immediately fixed with 4% PFA for 15 min at room temperature.

## Nocodazole MT recovery assay

MT regrowth assays were performed by treating cells with 3.3 $\mu$M nocodazole (Sigma-Aldrich) (or equal amount of DMSO) for 4 h at 37°C. Nocodazole washout was carried out three times with warm PBS and MTs were allowed to regrow in warm cell culture media (depending on the cell type) at 37°C for the desired time points. Before fixation, cells were rinsed with warm PHEM buffer (60 mM

PIPES, 25 mM HEPES, 10 mM EGTA, 4 mM $MgCl_2$), followed by an incubation step in PHEM buffer containing 0.05% Triton-X-100 at 37°C for 1.5 min to remove the soluble tubulin. Fixation was done with cold methanol (−20°C) for 8 min.

## EdU pulse labelling

Cells were incubated for a 24-h pulse in 0.2 $\mu$M EdU (component A of Click-iT EdU Imaging Kit; Invitrogen) at 37°C, 5% $CO_2$. After EdU incubation, cells were fixed in 3.7% formaldehyde in PBS for 15 min at RT. Formaldehyde was removed, and cells were washed twice with 1 ml of 3% BSA in PBS and permeabilized for 20 min at RT in 0.5% Triton X-100. EdU detection was performed according to the manufacturer's instructions, followed by immunostaining for other markers.

## Live imaging and EB3 comets tracking

NPCs were imaged at 37°C in an imaging buffer (170 mM NaCl, 3.5 mM KCl, 0.4 mM $KH_2PO_4$, 20 mM N-Tris[hydroxyl-methyl]-methyl-2-aminoethane-sulphonic acid [TES], 5 mM $NaHCO_3$, 5 mM glucose, 1.2 mM $Na_2SO_4$, 1.2 mM $MgCl_2$, 1.3 mM $CaCl_2$, pH 7.4) using Zeiss Axiovert 200 M microscope (Observer. Z1; Zeiss) equipped with 40X/1.40 Oil DIC objective; a pE-4000 LED light source (CoolLED) and a Hamamatsu Orca-Flash4.O V2 CMOS digital camera. Time-lapse images of DIV3 NPCs expressing EB3-Tdtomato and $\gamma$-Tubulin plasmids were acquired every second using Micro-Manager software (Micro-Manager1.4) for 20 s. EB3 comets tracking was performed using TrackMate (v6.0.1), an open-source Fiji (ImageJ) plugin (Tinevez et al, 2017), and using the EB3 tracking algorithm described in Meka et al (2022). As in the study by Meka et al (2022), we first preprocessed the EB3 time-lapse images before loading them on to TrackMate by substracting the background using the background tool in Fiji. Rolling ball radius selection of 10 pixels delineated the EB3 comets from the background distinctively. After loading the time-lapse to the TrackMate, LOG detector was selected, and the following parameters were given to be able to detect most but specific EB3 comets in the time-lapse: (i) blob diameter was chosen to ~0.45 microns, (ii) threshold value between 20 and 40 (based on the signal to noise ratio), (iii) median filter and sub-pixel localization parameters were set to on. In the next step, no initial thresholding was performed and continued to Hyperstack displayer, this step detects all the comets in the 20 frames. Next, to create tracks from the EB3 comets in the time-lapse, LAP tracker was selected, under which the following parameters were set without featuring any penalties: (a) for frame-to-frame linking of the comets track, maximum distance set was to 1 micron; (b) for track segment gap closing, gap closing was allowed and maximum distance set to 1 micron with a maximum frame gap set to 2; (c) track merging was allowed when the maximum distance was 1 micron. Using the analysis option all the data related to the EB3 comets and tracks were obtained as .csv files from which the following parameters were analyzed to study the EB3 dynamics: (1) EB3 track speed (microns per sec); (2) growth (displacement) of each EB3 comet per frame (microns) and (3) total number of each EB3 tracks (in seconds). To plot the EB3 tracks and comets dynamics near the centrosome, we used a custom Python code, where the XY

coordinates of the centrosome for each cell (obtained through Fiji–ImageJ by identifying the XY coordinates of the γ-Tubulin-marked centrosome) were set to (0, 0) and then the coordinates of the EB3 tracks XY coordinates (available in the data obtained after TrackMate analysis) were normalized accordingly.

24-h live imaging of neurospheres was performed using ImageXpress Micro 4 (Molecular Devices) equipped with a 40x Plan Apo Lambda, 0.95 NA. Cells were treated with SiR-DNA (Spirochrome-SC007) at a final concentration of 50 nM and imaged at 37°C, 5% $CO_2$, each 15 min for 24 h, after 1 h incubation.

### β-Gal senescence assay

The β-galactosidase staining in NPCs was done using the senescence β-galactosidase staining kit (#9860; Cell Signaling) following the manufacturer's guidelines. In brief, cells were fixed and incubated with 0.5 ml/35 mm well of the β-Galactosidase Staining Solution overnight at 37°C. The β-galactosidase-positive cells were considered senescent cells and counted in 10 randomly chosen fields per experiment and condition.

### Immunoblotting

NPCs were in radioimmunoprecipitation assay buffer (RIPA, 50 mM Tris pH 8.0, 150 mM NaCl, 1.0% IGEPAL CA-630, 0.5% sodium deoxycholate, 0.1% SDS) containing protease inhibitor (Roche) and phosphatase inhibitor (Thermo Fisher Scientific) using a cell scraper. Samples were incubated for 45 min on ice, followed by centrifugation (14,000$g$ for 20 min), and supernatant protein concentration assessment using Bradford assay (Sigma-Aldrich). 10–20 $\mu$g of samples were loaded on 10% SDS–PAGE gels, followed by transfer onto the nitrocellulose membrane. The membranes were blocked for 1 h at RT in 5% skim milk in TBS buffer (20 mM Tris pH = 7.6, 150 mM NaCl) containing 0.1% Tween (TBS-T) and incubated with primary antibodies (see Supplemental Data 3) overnight at 4°C. The next day, the membranes were incubated with a secondary HRP-tagged antibody (see Supplemental Data 3) for 1 h at RT. The membranes were washed and developed using an ECL-based autoradiography film system. Analysis was done using the Gel Analyzer plugin from Image J. For quantification, protein levels were always first normalized to the loading control, and then, the levels in the KO were normalized to the WT set to 100%.

### Co-immunoprecipitation

For immunoprecipitation experiments, 20 $\mu$l Dynabeads Protein G (Thermo Fischer Scientific) were coated with 2 $\mu$g antibody targeting the protein of interest and corresponding IgG as a negative control (see Supplemental Data 3). Therefore, Dynabeads storing solution was replaced by 100 $\mu$l PBS, and 2 $\mu$g of antibody was added. The beads were incubated with the antibody for 2–3 h at 4°C on a shaker before washing once with 200 $\mu$l PBS to remove excess antibodies. Embryos were isolated at E14–15 of pregnancy, and the telencephalic vesicles were extracted and homogenized in Co-IP buffer (50 mM Tris–HCl pH = 7.4, 100 mM NaCl, 1% NP-40, 2 mM MgCl2)

containing a protease inhibitor (Roche) and phosphatase inhibitor (Thermo Fisher Scientific). For the adult cortex, 8-wk-old mice were euthanized via cervical dislocation, and brains were isolated, dissected, and homogenized in Co-IP buffer using a Wheaton PotterElvehjem Tissue Grinder. Samples were sonicated and proteins were extracted for 45 min on ice, followed by the centrifugation of lysates at 16,200$g$ for 20 min at 4°C. Antibody-coupled Dynabeads were incubated with an equal amount of the supernatant overnight at 4°C on a shaker. The next day, Dynabeads were washed three times with Co-IP buffer before they were dissolved in 20 $\mu$l Co-IP buffer and 20 $\mu$l 4x SDS buffer and boiled at 95°C for 5 min. Precipitation of proteins was detected via SDS-page gel.

### Mass spectrometry analysis of AP-2α-binding partners

Embryos were obtained between stages E14–E16 of pregnancy and the telencephalic vesicles were extracted and homogenized in Co-IP buffer, as described in the previous section. After boiling the samples at 95°C for 5 min, the samples were loaded onto SDS–PAGE gels, reduced (DTT), and alkylated (CAA). Digestion was performed using trypsin at 37°C overnight. Peptides were extracted and purified using Stagetips. Eluted peptides were dried in vacuo, resuspended in 1% formic acid/4% acetonitrile, and stored at –20°C before MS measurement. All samples were analyzed by the CECAD proteomics facility on a Q Exactive Plus Orbitrap mass spectrometer that was coupled to an EASY nLC (both Thermo Fisher Scientific). Peptides were loaded with solvent A (0.1% formic acid in water) onto an in-house–packed analytical column (50 cm, 75 $\mu$m I.D., filled with 2.7 $\mu$m Poroshell EC120 C18; Agilent). Peptides were chromatographically separated at a constant flow rate of 250 nl/min using the following gradient: 3–7% solvent B (0.1% formic acid in 80% acetonitrile) within 1.0 min, 7–23% solvent B within 35.0 min, 23–32% solvent B within 5.0 min, 32–85% solvent B within 5.0 min, followed by washing and column equilibration. The mass spectrometer was operated in data-dependent acquisition mode. The MS1 survey scan was acquired from 300–1,750 m/z at a resolution of 70,000. The top 10 most abundant peptides were isolated within a 1.8 Th window and subjected to HCD fragmentation at a normalized collision energy of 27%. The AGC target was set to 5 × 10$^5$ charges, allowing a maximum injection time of 108 ms. Product ions were detected in the Orbitrap at a resolution of 35,000. Precursors were dynamically excluded for 20.0 s. All mass spectrometric raw data were processed with Maxquant (version 1.5.3.8, [Tyanova et al, 2016a]) using default parameters. Briefly, MS2 spectra were searched against the UniProt mouse reference proteome containing isoforms (UP589, downloaded on 26.08.2020), including a list of common contaminants. False discovery rates on protein and PSM level were estimated by the target–decoy approach to 1% (Protein FDR) and 1% (PSM FDR), respectively. The minimal peptide length was set to 7 amino acids and carbamidomethylation at cysteine residues was considered as a fixed modification. Oxidation (M), Phospho (STY), and Acetyl (Protein N-term) were included as variable modifications. The match-between-runs option was enabled between replicates. LFQ quantification was enabled using default settings. Follow-up analysis was done in Perseus 1.6.15 ([Tyanova et al, 2016b]).

## Proteomics

NPCs were lysed in 8 M Urea/50 mM TEAB buffer containing protease inhibitors. Samples were sonicated, followed by the centrifugation of lysates at 20,000$g$ for 15 min. Lysates were reduced (DTT) and alkylated (CAA). Digestion was performed using trypsin at 37°C overnight. Peptides were extracted and purified using Stagetips. All samples were analyzed by the CECAD proteomics facility on a Q Exactive Plus Orbitrap mass spectrometer that was coupled to an EASY nLC (both Thermo Fisher Scientific). Peptides were loaded with solvent A (0.1% formic acid in water) onto an in-house packed analytical column (50 cm — 75 $\mu$m I.D., filled with 2.7 $\mu$m Poroshell EC120 C18; Agilent). Peptides were chromatographically separated at a constant flow rate of 250 nl/min using the following gradient: 3–4% solvent B (0.1% formic acid in 80% acetonitrile) within 1.0 min, 5–27% solvent B within 119.0 min, 27–50% solvent B within 19.0 min, 50–95% solvent B within 1.0 min, followed by washing and column equilibration. The mass spectrometer was operated in data-dependent acquisition mode. The MS1 survey scan was acquired from 300–1,750 m/z at a resolution of 70,000. The top 10 most abundant peptides were isolated within a 1.8 Th window and subjected to HCD fragmentation at a normalized collision energy of 27%. The AGC target was set to 5 × 10$^5$ charges, allowing a maximum injection time of 55 ms. Product ions were detected in the Orbitrap at a resolution of 17,500. Precursors were dynamically excluded for 30.0 s. All mass spectrometric raw data were processed with Maxquant (version 1.5.3.8, [Tyanova et al, 2016a]) using default parameters against the UniProt canonical mouse database (UP589, downloaded on 15.08.2019) with the match-between-runs option enabled between replicates. Follow-up analysis was done in Perseus 1.6.15 (Tyanova et al, 2016b).

## Flow cytometry

NPCs were harvested, centrifuged (100$g$, 3 min), and washed with 1 ml PBS. Half of the PBS was removed and 9.5 ml ice-cold 70% ethanol was added to the cells dropwise for fixation. Cells were spined down at 1,400$g$ before staining to remove ethanol and washed with PBS twice. The cell pellet was resuspended in a propidium iodide solution (Pl 10 $\mu$g/ml, RNAse A 200 $\mu$g/ml, and 0.1% TritonX-100 in 1xPBS) and incubated 5 min at 37°C or 30 min at RT. Flow cytometry was performed using BD LSR Fortessa (BD), gating the analysis to 20,000 events (or in the case of KO cells because of the low number till the sample was thoroughly analyzed) and analyzed using Flowing software 2. pMEF cells were collected by using trypsin and centrifuged at 1,000$g$ for 3 min. Cells were counted and continued with ~ 1 × 106 cells. Then, they were washed with 1 ml 1xPBS and centrifuged again at 1,000$g$ for 3 min. Half of the PBS was removed and 9.5 ml of ice-cold 70% ethanol was added to the cells dropwise on vortex for fixation. Cells were spun down at 1,400$g$ for 3 min before staining to remove ethanol and washed with PBS twice. After the last wash, cells were spun down again and PBS removed. 0.5 ml of FxCycle PI/RNase Staining Solution (F10797; Thermo Fisher Scientific) was added to each flow cytometry sample to stain and mixed well.

## Migration assay

Neurospheres were pipetted without dissociation and seeded in Matrigel-coated coverslips. After 6, 12, 24, and 48 h, cells were observed and imaged in a brightfield microscope before fixation. Immunostaining was performed as described above. Migrated distance and the number of migrating cells were analyzed in ImageJ.

## Microscopy

Fixed cells were imaged using Zeiss Axiovert 200 M microscope (Observer. Z1; Zeiss) equipped with 40x/1.4 oil DIC objective and 63×/1.4 oil DIC objective and the Micro-Manager software and/or with Zeiss LSM Meta 710 confocal microscope equipped with 405, 488, 543, and 633 nm laser lines and Plan-Apochromat 63x/1,4 Oil DIC objective. Coverslips were imaged using EVOS FL Auto 2 (Invitrogen) to quantify the number of senescent cells. For quantitative analysis, several pictures were considered until quantification of a significant number of cells per experiment. Fluorescent protein levels were analyzed by manually selecting the cell body using ImageJ selection tools (ROI), and the mean gray value was quantified within the ROI after the background subtraction. Background substation was performed by selecting the ROI within the image field of view not containing cells. The cell boundary was defined using Vimentin co-immunostaining as a mask, and/or using phase contrast images of the cells. CEP135 and/or $\gamma$-Tubulin immunostaining were used to define the ROI for centrosomal localization of GCP2 and GCP3. All imaging parameters were kept constant among related experiments.

## Statistical analysis

Statistical analyses were done on cell values (indicated by data points) from at least three independent experiments and biological replicates (except for Figs 3C and S4E). MS Excel (Microsoft) and GraphPad Prism version 7 (GraphPad Software, Inc.) were used for statistical analysis and result illustration. Normalized data between two groups were analyzed using a one-tailed unpaired $t$ test. Statistical difference between more than two groups were compared with one-way ANOVA (Tukey's posthoc test was used to determine the statistical significance between the groups). Statistical difference between more than two groups and two conditions was evaluated using two-way ANOVA (Holm–Sidak post-hoc test was used to determine the statistical significance between the groups). Statistical difference of cell cycle stages distribution within WT and KO cells was evaluated using $x^2$ test using original cell counts. Significant differences were accepted at $P < 0.05$ indicated by asterisks: $*P < 0.05$; $**P < 0.01$; $***P < 0.001$, $****P < 0.0001$.

# Data Availability

All data needed to evaluate the conclusions in the article are present in the article and/or the Supplementary Materials. Proteome data of all experiments are deposited in the database PRIDE

and accessible for public after publishing (Project accession numbers: PXD047200 and PXD047202). Additional data related to this article may be requested from the corresponding author.

# Supplementary Information

# Acknowledgements

We thank Marvin Schäfer and Sylvia Müller for expert technical assistance. We are indebted to Dr. Christian Jüngst (CECAD Imaging Facility), Dr. Stephan Müller, and Dr. Jan-Wilm Lackmann (CECAD Proteomic Facility) for their help and expert assistance. We are indebted to Dr. Christian Mönch (FZ Jülich) for his help with the custom Python code for analysis of EB3 comet tracks. The work of NL Kononenko is funded by the Deutsche Forschungsgemeinschaft (DFG, German Research Foundation): EXC 2030–390661388, KO 5091/4-1, DFG-431549029–SFB 1451, and DFG-411422114–GRK 2550.

## Author Contributions

S Camblor-Perujo: formal analysis and investigation.
E Ozer Yildiz: formal analysis, investigation, and visualization.
H Küpper: investigation.
M Overhoff: investigation and visualization.
S Rastogi: formal analysis and investigation.
H Bazzi: conceptualization, resources, and supervision.
NL Kononenko: conceptualization, data curation, supervision, funding acquisition, investigation, visualization, and writing—original draft, review, and editing.

## Conflict of Interest Statement

The authors declare that they have no conflict of interest.

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
