## [Reviewer comments · Life Science Alliance]

Life Science Alliance

The AP-2 complex interacts with γ TuRC and regulates the proliferative capacity of neural progenitors

Santiago Cambor-Perujo, Ebru Özer Yildiz, Hanna Küpper, Melina Overhoff, Saumya Rastogi, Hisham Bazzi, and Natalia Kononenko

DOI: <https://doi.org/10.26508/lsa.202302029>

Corresponding author(s): Natalia Kononenko, Center for Physiology and Pathophysiology

Review Timeline:

Submission Date:	2023-03-07
Editorial Decision:	2023-04-24
Revision Received:	2023-10-20
Editorial Decision:	2023-11-16
Revision Received:	2023-11-23
Accepted:	2023-11-27

Transaction Report:

April 24, 2023

Re: Life Science Alliance manuscript #LSA-2023-02029-T

Prof. Natalia L. Kononenko
CECAD, University of Cologne
Faculty of Medicine and University Hospital Cologne, University of Cologne
Joseph-Stelzmann-Str. 26
Cologne, NRW 50931
Germany

Dear Dr. Kononenko,

Thank you for submitting your manuscript entitled "The endocytic adaptor AP-2 interacts with γ -TuRC components and regulates cell cycle progression of neural progenitor cells" to Life Science Alliance. The manuscript was assessed by expert reviewers, whose comments are appended to this letter. We invite you to submit a revised manuscript addressing the Reviewer comments.

Thank you for this interesting contribution to Life Science Alliance. We are looking forward to receiving your revised manuscript.

Sincerely,

B. MANUSCRIPT ORGANIZATION AND FORMATTING:

Reviewer #1 (Comments to the Authors (Required)):

This manuscript from Natalia Kononenko's group proposes a novel, non-canonical function for AP-2 (the main adaptor protein for clathrin-mediated endocytosis in mammals) in regulating cell cycle progression of neural progenitor cells. Based on the conditional knockout/knockdown phenotypes of AP-2 μ in mouse neurons, the authors explain the observed defects in centrosome formation and mitotic progression, accompanied by DNA damage and accumulation of p53. The authors then show that AP-2 binds to γ tubulin RC components and localizes to centrosomes in order to regulate cell cycle possibly via MT nucleation. Further, they show that this function of AP-2 is independent of its canonical role in clathrin-mediated endocytosis. Overall, the study is interesting in that it suggests novel functions for AP-2 at a physiological level which is of interest to cell/developmental/ neuro biologists. The data is more descriptive (not mechanistic) and methodology used is rigorous. The manuscript is well laid out. However, I feel that the following points must be addressed to make it more suitable for publication in Life Science Alliance.

1) AP-2 knockdown/knockout is expected to cause multiple effects (direct or indirect) in the system. The same group has shown distinct AP-2 phenotypes in neurons in the past. Hence, to strongly link AP-2 function to the phenotype observed, a functional rescue experiment with AP-2 μ is highly appreciated.

2) The clathrin-independent role of AP-2 in cell proliferation is not very clear. The authors have used a pharmacological approach (Pitstop inhibitor) to suppress clathrin function and compared the centrosome phenotypes with AP-2 conditional knockouts/knockdowns created via genetic approach. These comparisons largely depend on varying degrees of functional ablation of proteins. Suppressing clathrin function in the same system using another strategy if possible eg, knockdown would support the inhibitor experiment. More importantly, the endocytic defects after ablating AP-2 or clathrin would act as an internal control for comparing the phenotypes.

3) Given that AP-2 is the central adaptor protein for clathrin-mediated endocytosis, it is surprising to see that in this study the authors have not shown any endocytic phenotypes for AP-2 knockout/knockdown in neurons. This need to be addressed experimentally.

Reviewer #2 (Comments to the Authors (Required)):

Camblo-Perujo et al. studied the effects of disruption of a key clathrin-mediated endocytosis (CME) adaptor complex, AP-2, on primary neuronal progenitor cells (NPCs) and mouse embryonic fibroblasts (MEFs). The authors used an engineered mouse line allowing tamoxifen (Tmx)-driven recombination knocking-out (KO) AP-2 central μ subunit (Ap2m1), which destabilizes the whole AP-2 complex. The authors wisely aimed at separating the CME-dependent and -independent effects of AP-2 on NPCs proliferation and differentiation. Using well-done global proteomic analysis, co-immunoprecipitation and live imaging, Camblo-Perujo et al. uncover that AP-2 interacts with γ -Tubulin Ring Complex (γ -TuRC), a key catalyzer of microtubule (MT) nucleation, that is found concentrated on MT organizing centers such as the centrosome. The finding is of great interest since several previous works implicated members of endocytic pathways in the regulation of MT-driven processes, which suggests a novel functional cross-talk between endocytic and microtubule pathways.

By series of imaging experiments, the authors show that AP-2 destruction disrupts centrosomal integrity, centrosomal MT nucleation, and mitotic spindle formation and function, which leads to cell cycle arrest and cell death. Concomitantly, the surviving population of AP-2 μ KO NPCs exhibits accelerated neuronal differentiation. The authors seem to want to associate the proliferation defects of AP-2 KO cells with previously described "centrosome surveillance" pathway linked to the tumor suppressor p53. To this end, they prepared immortalized MEFs (iMEFs), that were able to overcome the mitotic block observed in AP-2 μ KO NPCs and MEFs. Unfortunately, the experiments on MEFs are incomplete, as detailed below, and additional data are required to support the conclusions of the manuscript.

Overall, it is a promising study describing a novel regulation of γ -TuRC-mediated MT nucleation via AP-2, which could significantly advance our knowledge on the mechanisms of the centrosome assembly and function and the impact of these processes on the regulation of cell cycle and cell death. However, the manuscript suffers from several experimental and conceptual shortcomings that must be experimentally addressed.

Major points:

1. The global proteomic analysis is well performed, the data are strongly supportive. The cell proliferation data look also fine. However, the authors need to explain the discrepancy between their cell cycle analysis shown in Fig. 1M and in Fig. 4F-4G. Specifically, the proportion of alive mitotic cells (i.e. mitotic index) in AP-2 μ KO seems to be lower compared to the control in Fig. 1M; instead, Figs. 4F-4G show a clear increase in the mitotic index.
2. Co-immunoprecipitation experiments are well performed, but, reciprocal co-immunoprecipitation is needed, using a γ -TuRC component as a bait. We recommend using GCP2-01 monoclonal antibody that works great in co-immunoprecipitation experiments using E18 mouse embryonic cortices.
3. Pitstop2 titration and clear evidence for CME abolishment at 30 μ M concentration used in the paper in NPCs and MEFs/iMEFs must be shown, since it is a different system than in the referenced works.
4. Additional experiments need to be performed to clearly show the fate of γ -TuRC components in AP-2 μ KO NPCs. All claims about the decrease of γ -TuRC component (γ -tubulin, GCP2 or GCP3, and GCP4) protein level need to be documented by Western blotting (WB).
5. The immunofluorescence (IF) analysis of subcellular distribution of γ -TuRC components must focus on the centrosomal localization. Together with the studied γ -TuRC components, the authors should also mark the centrosome by a reliable marker (see below) stably present in both wt and KO NPCs. The absence of the γ -TuRC components at the centrosome might be related to delocalization rather than to protein degradation.
6. The presence, absence or multiplication of the centrosome in AP-2 μ KO NPCs is very interesting. However it needs to be shown clearly by staining for core centriolar proteins or inner layer PCM proteins. We recommend using Cep135 (ab75005) or Cep120, preferentially proteins that are not trafficked through centriolar satellites (based on the phenotype, authors might want to consult: <https://journals.biologists.com/jcs/article/133/1/jcs239566/224728/Centriolar-satellite-biogenesis-and-function-in-;> or newer literature).
7. Microtubule regrowth experiments are not performed well. It is therefore impossible to interpret them correctly. First, the fixation procedure did not work well, even the untreated controls show disrupted and overextracted MT network (Fig. 3G, EV3A). Instead of an atypical microtubule regrowth procedure described in Methods, please refer to classical MT regrowth protocols employed in recent papers, e.g. in Camargo Ortega, 2019; <https://www.nature.com/articles/s41586-019-0962-4>, where it was also used for NPCs.
Moreover, the nocodazole treatment was probably insufficient and is described incorrectly. The stated 0.2 μ g/ml concentration in the Methods is not the 5 μ M concentration stated in the figure legend. It is almost 10 times lower. The authors should find the nocodazole concentration and treatment duration to achieve complete MT depolymerization.
8. A serious problem of the EB3 experiments is very high EB3 overexpression. The observed "static" behaviour might be an artifact (when highly overexpressed, EB3 binds to whole MT that would "freeze"). We recommend either substantial lowering of the expression of the current plasmid or using another vector (EB3-mNeonGreen probably the best). The authors should be able to visualize EB3 as MT plus end comets exhibitin none or only minimal binding of EB3 along the length of MT. Moreover, the movie showing the described and quantified (Fig. 3I) "static EB3" phenotype is missing in the Supplement and should be shown. Further, to be able to claim that something is centrosomal, the centrosome must be shown - the authors would need to find a stable marker (see point 6). Alternatively, if authors would clearly show that the whole centrosome decomposes in AP-2 μ KO NPCs, the graph description in Fig. 3J should be changed to "Perinuclear MT asters" or something similar.
9. The experiments in MEFs could nicely complement the first part of the study, however, many important complementary experiments are missing which is not comprehensible. Also the language in the text describing the results is not clear. Analysis of γ -TuRC components and the centrosome integrity experiments suggested above should be done also on MEFs and iMEFs (always wt vs. AP-2 μ KO), including correctly performed MT regrowth experiments. In addition, there is no quantification of the experiment shown in Fig. EV5E and it should be complemented by the same experiment done on "normal" MEFs (wt vs. KO). Further, analysis of centrosome maturation (a centrosome marker plus γ -tubulin) should be done on MEFs and iMEFs (always wt vs. KO), comparing interphase vs. G2/M (pHH3-positive) cells. Quantification of the Fig. 5J regarding cell number ["# of iMEFs (% of WT)] need to be shown. Cell cycle analysis (FACS) would be beneficial for both pMEFs and iMEFs (always wt vs. KO) as well. All data must be then interpreted according to new results. Rather than manipulating G1/S, it seems that the authors might have overridden the G2/M and mitotic checkpoints.

Minor points:

1. Evidence for AP-2 knock-out efficiency is strong by analogy and reference to the previous work of the authors. However, in a new system, it should also be shown in DIV3 tamoxifen-treated NPCs using the anti-AP-2 μ antibody (which is apparently working, since it was used for another experiment in Fig. EV2A).
2. Line 46 -The sentence is ambiguous and needs to be rephrased. In our opinion, "The proliferative capacity of NPCs is determined..." mostly by cell-intrinsic factors which then dictate the number of division rounds and the varying identity of daughter cells. And not vice-versa.
3. Line 55 - Centrosomes might actually be enclosed by a membrane, at least in *Caenorhabditis elegans* (<https://www.sciencedirect.com/science/article/abs/pii/S0960982222019960>).
4. Line 61 - Accompany the citation of Schiebel, 2000, by newer reviews; taking into account for example newer findings on CKAP5 and γ -TuRC.

5. Line 93-95 - The sentence and the citations are imprecise. Some of the cited works are about non-centrosome/spindle/midbody proteins regulating the mitosis actually through membrane morphology (e.g. Epsin1). Thus, the sentence should be rephrased (divide clearly canonical vs. non-canonical effects of CME components on the described processes and also distinguish clearly canonical and non-canonical/novel localization of the discussed CME components) and add only relevant citations added. Some of the cited works are not linked to the topic directly (e.g. He et al, 2002); and this could be confusing for readers outside of the field.
6. Use γ -tubulin instead of GCP1 (Figure EV2)- it is an uncommon term and could be mistakenly associated with another protein. Also the term "astral" microtubules is classically given to the MTs emanating for mitotic spindle poles (mitotic centrosomes) towards the cell membrane. It seems that the authors use this term also for interphase cells. We would recommend using the term "centrosomal microtubules" for MTs emanating from the centrosome in the interphase.
7. Check for typos throughout the manuscript (AP2- μ instead of AP-2 μ in Fig. EV2A, Line 57 - animals cells, Line 297 - Fig. 5M, Fig. EV5E legend: iMEFs (not NPCs), Line 840 - missing spaces etc.)
8. Show GCP3 and AP-2 μ colocalization in the Supplement (Missing pictures for the data in the graph in Fig. 2E)
9. Always use "neuronal progenitors" and not only "neuronal" when you are describing data acquired on NPCs. "Neuronal proliferation" - lines 277 and 279 is generally incorrect.
10. Line 307 - "where it interacts with components..." This is not so straightforward. It is likely; however, the definite proof would be to see both γ -TuRC and AP-2 on isolated centrosomes or at least in a centrosomal fraction (without cytoplasmic complexes).
11. Line 308 - MT anchoring was not addressed at all. MT nucleation is likely true based on the data, however, MT regrowth experiments and EB3 live imaging must be redone correctly (see above).
12. Line - 348 "centrosome overduplication" - There are no data shown to support this. The authors checked only γ -tubulin staining. The major phenotype was "centrosome loss" (Fig. 2M), meaning γ -tubulin double-dot centrosomal signal was lost. The multiple centrosomes observed in the rest of the cells could be γ -tubulin cytoplasmic puncta that can form under various conditions as shown by others before. However, the puncta are usually no centrosomes (they lack centrioles). More experiments would be needed to address this properly (i.e. always showing a reliable centriolar counterstain to distinguish multiple centrosomes vs. acentrosomal puncta of centrosomal proteins)
13. Line 366 - "neuronal embryonic" - better: embryonic neuronal progenitor cells
14. Line 368 - "of the γ -TuRC" - better: "of the active γ -TuRC"
15. Lines 377-378 - "... whereby AP-2 regulates centrosome function at the level of MT nucleation by recruiting components of the γ -TuRC complex." - Rephrasing needed. The data rather show that AP-2 is required for γ -TuRC stability, however, additional experiments, suggested above, are needed to establish this (missing WBs - wt vs. AP-2 KO). No data regarding γ -TuRC centrosomal recruitment through AP-2 shown.
16. Lines 388-389 - "AP-2 may contribute to the expansion of pericentriolar material at the onset of mitosis by recruiting additional γ -TuRC complexes, thereby increasing MT nucleation activity as the cell cycle progresses." - Not shown, but easy to show by comparing interphase and mitotic centrosomes in wt vs. KO as suggested above
17. Line 433 - use relative centrifugation force (rcf) rather than rpm
18. Line 648 - typo - Pitstop2
19. Line 498 - "NPCs co-expressing (X)" - Unclear what does the X mean?
20. Line 607 - "...by manually selecting the cell body using ImageJ selection tools (ROI)," - How was the cell boundary assessed? How was the background subtraction carried out? Were the imaging parameters kept constant among related experiments? This all should be stated. Immunofluorescence analysis should also focus directly on centrosomes as mentioned above.
21. The authors should consult a statistician for several analyses. For Fig. 1L,M,N,O, Fig. 2M, O, the t-test is not correct. A chi-square test might be more appropriate: <https://www.sciencedirect.com/science/article/pii/S001216061001016X>
Figure EV3H,I - the data seem to be longitudinal, repeated measures ANOVA might be more appropriate.
22. Timer is missing in all movies and scale bars in EB3 movies. A starting slide describing the experiment would be good for all the movies. A centrosomal marker should be shown (even if the complete centrosome decomposition is confirmed in AP-2 KO, the control should be clear).

Point by Point Response to Reviewers.

We would like to thank the reviewers for their constructive comments. Below, we address all remaining concerns that were expressed by reviewers.

Reviewer #1 (Comments to the Authors (Required)):

This manuscript from Natalia Kononenko's group proposes a novel, non-canonical function for AP-2 (the main adaptor protein for clathrin-mediated endocytosis in mammals) in regulating cell cycle progression of neural progenitor cells. Based on the conditional knockout/knockdown phenotypes of AP-2 μ in mouse neurons, the authors explain the observed defects in centrosome formation and mitotic progression, accompanied by DNA damage and accumulation of p53. The authors then show that AP-2 binds to γ tubulin RC components and localizes to centrosomes in order to regulate cell cycle possibly via MT nucleation. Further, they show that this function of AP-2 is independent of its canonical role in clathrin-mediated endocytosis. Overall, the study is interesting in that it suggests novel functions for AP-2 at a physiological level which is of interest to cell/ developmental/ neuro biologists.

We thank reviewer 1 for this positive comment on the novelty of our work.

The data is more descriptive (not mechanistic) and methodology used is rigorous. The manuscript is well laid out. However, I feel that the following points must be addressed to make it more suitable for publication in Life Science Alliance.

We appreciate the reviewer's comment on the rigor of our methodology and the fact that our manuscript is well designed. We have now addressed all of the points raised by the reviewer as listed below.

1) AP-2 knockdown/knockout is expected to cause multiple effects (direct or indirect) in the system. The same group has shown distinct AP-2 phenotypes in neurons in the past. Hence, to strongly link AP-2 function to the phenotype observed, a functional rescue experiment with AP-2 μ is highly appreciated.

We thank the reviewer for this comment. We have now gladly followed the reviewer's suggestion and performed a functional rescue experiment using the plasmid overexpressing AP-2 μ (AP-2 μ -IRES-mRFP). As it can be appreciated from the new data in Fig. 2O (and Fig. EV2L), overexpression of AP-2 μ rescues the centrosome morphology phenotype in AP-2 KO NPCs.

2) The clathrin-independent role of AP-2 in cell proliferation is not very clear. The authors have used a pharmacological approach (Pitstop inhibitor) to suppress clathrin function and compared the centrosome phenotypes with AP-2 conditional knockouts/knockdowns created via genetic approach. These comparisons largely depend on varying degrees of functional ablation of proteins. Suppressing clathrin function in the same system using another strategy if possible eg, knockdown would support the inhibitor experiment. More importantly, the endocytic defects after ablating AP-2 or clathrin would act as an internal control for comparing the phenotypes.

We thank the reviewer for this comment. We have performed a series of experiments where we suppressed clathrin function via shRNA-mediated knockdown of CHC. As it can be appreciated from the new data in Fig. EV2Q,P, knockdown of CHC has no significant effect on centrosome morphology in WT NPCs, supporting data obtained in NPCs treated with Pitsstop2. Furthermore, we have compared endocytic phenotypes (via uptake of transferrin) in NPCs lacking AP-2 μ with NPCs where clathrin function was suppressed with either Pitsstop2 or CHC shRNA-mediated knockdown. In all three conditions, uptake of transferrin is strongly and comparably inhibited (please see new data in Fig. EV1 C,D, Fig. EV1 H,I and Fig. EV2 O,P).

3) Given that AP-2 is the central adaptor protein for clathrin-mediated endocytosis, it is surprising to see that in this study the authors have not shown any endocytic phenotypes for AP-2 knockout/knockdown in neurons. This need to be addressed experimentally.

We have now performed uptake of Alexa 488-labeled transferrin in NPCs lacking AP-2 μ . As it can be appreciated from the new data in Fig. EV1C,D, AP-2 μ -deficient NPCs are strongly impaired in their ability to internalize transferrin. This confirms the endocytic phenotype of AP-2 KO NPCs. We would like to note that the endocytic (CME) and additional non-canonical phenotype of AP-2 KO neurons were analyzed in detail in our previous papers (Kononenko et al., 2014, Kononenko et al., 2017, Bera et al., 2020).

Reviewer #2 (Comments to the Authors (Required)):

Camblo-Perujo et al. studied the effects of disruption of a key clathrin-mediated endocytosis (CME) adaptor complex, AP-2, on primary neuronal progenitor cells (NPCs) and mouse embryonic fibroblasts (MEFs). The authors used an engineered mouse line allowing tamoxifen (Tmx)-driven recombination knocking-out (KO) AP-2 central μ subunit (Ap2m1), which destabilizes the whole AP-2 complex. The authors wisely aimed at separating the CME-dependent and -independent effects of AP-2 on NPCs proliferation and differentiation. Using well-done global proteomic analysis, co-immunoprecipitation and live imaging, Camblo-Perujo et al. uncover that AP-2 interacts with γ -Tubulin Ring Complex (γ -TuRC), a key catalyzer of microtubule (MT) nucleation, that is found concentrated on MT organizing centers such as the centrosome. The finding is of great interest since several previous works implicated members of endocytic pathways in the regulation of MT-driven processes, which suggests a novel functional cross-talk between endocytic and microtubule pathways.

We thank the reviewer 2 for this positive comment on our work.

By series of imaging experiments, the authors show that AP-2 destruction disrupts centrosomal integrity, centrosomal MT nucleation, and mitotic spindle formation and function, which leads to cell cycle arrest and cell death. Concomitantly, the surviving population of AP-2 μ KO NPCs exhibits accelerated neuronal differentiation. The authors seem to want to associate the proliferation defects of AP-2 KO cells with previously described "centrosome surveillance" pathway linked to the tumor suppressor p53. To this end, they prepared immortalized MEFs (iMEFs), that were able to overcome the mitotic block observed in AP-2 μ KO NPCs and MEFs.

Unfortunately, the experiments on MEFs are incomplete, as detailed below, and additional data are required to support the conclusions of the manuscript.

Overall, it is a promising study describing a novel regulation of γ -TuRC-mediated MT nucleation via AP-2, which could significantly advance our knowledge on the mechanisms of the centrosome assembly and function and the impact of these processes on the regulation of cell cycle and cell death. However, the manuscript suffers from several experimental and conceptual shortcomings that must be experimentally addressed.

We appreciate the reviewer's positive comment on the novelty of this study. We have now performed a series of new experiments, where we addressed reviewer's concerns. The outcome of these experiments supports the conclusions of our original manuscript. Especially, we performed a rigorous analysis of pMEFs and iMEFs in terms of centrosome morphology and γ -TuRC components localization.

Major points:

1. The global proteomic analysis is well performed, the data are strongly supportive. The cell proliferation data look also fine. However, the authors need to explain the discrepancy between their cell cycle analysis shown in Fig. 1M and in Fig. 4F-4G. Specifically, the proportion of alive mitotic cells (i.e. mitotic index) in AP-2 μ KO seems to be lower compared to the control in Fig. 1M; instead, Figs. 4F-4G show a clear increase in the mitotic index.

We thank the reviewer for this comment. We believe that the difference in the data lies in the fact that FACS analysis assesses the G2/M phases combined, whereas in the immunostaining we specifically looked for mitotic cells (M-phase alone). We have now analyzed the G2 cells (using its characteristic pHH3 pattern, which is distinct from that in M-phase) and found that the number of G2 cells is significantly decreased in AP-2 KO NPCs. In fact, the graph presented in Fig. EV4B (the G2/M cells in WT and KO) is comparable with the data shown in Fig. 1M. Our hypothesis is that KO NPCs are undergoing apoptosis after aberrant mitosis.

2. Co-immunoprecipitation experiments are well performed, but, reciprocal co-immunoprecipitation is needed, using a γ -TuRC component as a bait. We recommend using GCP2-01 monoclonal antibody that works great in co-immunoprecipitation experiments using E18 mouse embryonic cortices.

We have now made a strong effort in performing several reciprocal co-immunoprecipitation experiments, where GCP2 (using the antibody recommended by the reviewer) or GCP3 were used as a bait. We have observed that γ -TuRC components can also associated with AP-2 complex, albeit with lower efficiency (Fig. EV2 B,C).

3. Pitstop2 titration and clear evidence for CME abolishment at 30 μ M concentration used in the paper in NPCs and MEFs/iMEFs must be shown, since it is a different system than in the referenced works.

We have now analyzed CME inhibition (via uptake of transferrin) in NPCs treated with 30 μ M of Pistop2 for 24h and 48h. We found that uptake of transferrin is strongly inhibited under this condition. More importantly, the inhibition of CME is exactly comparable with the endocytic phenotype of AP-2 μ KO NPCs (please see new data in Fig. EV1 C,D, Fig. EV1 H,I). We would like to note that we did not use Pitstop2 in pMEFs and/or iMEFs.

4. Additional experiments need to be performed to clearly show the fate of γ -TuRC components in AP-2 μ KO NPCs. All claims about the decrease of γ -TuRC component (γ -tubulin, GCP2 or GCP3, and GCP4) protein level need to be documented by Western blotting (WB).

We thank the reviewer for this comment. We would like to note though that the collection of lysates from AP-2 KO NPCs is very challenging due to their decreased proliferation and increased mortality (the protein amount is always very low in KO conditions and requires pooling of several embryos, which is not always possible due to the genotypes per litter). Nevertheless, we have now made a significant effort to address reviewer's concern and present the WB analysis of GCP2 in AP-2 KO NPCs in Fig. EV2E. As it can be appreciated by the reviewer, the levels of GCP2 are significantly reduced in AP-2 KO NPCs.

5. The immunofluorescence (IF) analysis of subcellular distribution of γ -TuRC components must focus on the centrosomal localization. Together with the studied γ -TuRC components, the authors should also mark the centrosome by a reliable marker (see below) stably present in both wt and KO NPCs. The absence of the γ -TuRC components at the centrosome might be related to delocalization rather than to protein degradation.

We thank the reviewer for this comment. We have now performed a series of new experiments, where we analyzed subcellular distribution of GCP2 and GCP3 specifically focusing on centrosome, marked by CEP135 (and/or γ -Tubulin). Our new data indicate that the levels of GCP2 and GCP3 are reduced at the centrosome in NPCs, pMEFs and iMEFs. We also found that there is a significant decrease in the percentage of total overlap between GCP2, GCP3 and centrosomal markers. These data support our conclusion in the original manuscript and indicate that AP-2 is required for the localization of γ -TuRC components at the centrosome. We agree with the reviewer that at the current stage, we don't know if e.g. loss of GCP2 levels (Fig. EV2E) is due to its protein degradation. We have now modified the text in the manuscript to avoid this misconception.

6. The presence, absence or multiplication of the centrosome in AP-2 μ KO NPCs is very interesting. However it needs to be shown clearly by staining for core centriolar proteins or inner layer PCM proteins. We recommend using Cep135 (ab75005) or Cep120, preferentially proteins that are not trafficked through centriolar satellites (based on the phenotype, authors might want to consult: <https://journals.biologists.com/jcs/article/133/1/jcs239566/224728/Centriolar-satellite-biogenesis-and-function-in>; or newer literature).

We thank the reviewer for this comment and the suggestion to confirm our results with a core centriolar protein. We have now performed a rigorous analysis of centrosome morphology using co-immunostaining for CEP135 (as suggested by the reviewer) together

with γ -Tubulin in WT and KO NPCs. Our new data indicate that the centrosome morphology (marked by CEP135) is impaired in AP-2 KO NPCs and support the original conclusion of loss of centrosome integrity in NPCs lacking AP-2.

7. Microtubule regrowth experiments are not performed well. It is therefore impossible to interpret them correctly. First, the fixation procedure did not work well, even the untreated controls show disrupted and overextracted MT network (Fig. 3G, EV3A). Instead of an atypical microtubule regrowth procedure described in Methods, please refer to classical MT regrowth protocols employed in recent papers, e.g. in Camargo Ortega, 2019; <https://www.nature.com/articles/s41586-019-0962-4>, where it was also used for NPCs. Moreover, the nocodazole treatment was probably insufficient and is described incorrectly. The stated 0.2 μ g/ml concentration in the Methods is not the 5 μ M concentration stated in the figure legend. It is almost 10 times lower. The authors should find the nocodazole concentration and treatment duration to achieve complete MT depolymerization.

We thank the reviewer for this comment. We have re-done the microtubule re-growth experiments using the protocol employed in Camargo Ortega et al., 2019. We have now achieved complete MT depolymerization using 3,3 μ M nocodazole applied for 4 h and marked the centrosome by CEP135 immunostaining. Our new analyses indicate that AP-2 KO NPCs, although not impaired in bulk MT regrowth after their release from nocodazole, grow significantly less centrosomal MTs.

8. A serious problem of the EB3 experiments is very high EB3 overexpression. The observed "static" behaviour might be an artifact (when highly overexpressed, EB3 binds to whole MT that would "freeze"). We recommend either substantial lowering of the expression of the current plasmid or using another vector (EB3-mNeonGreen probably the best). The authors should be able to visualize EB3 as MT plus end comets exhibit in none or only minimal binding of EB3 along the length of MT. Moreover, the movie showing the described and quantified (Fig. 3I) "static EB3" phenotype is missing in the Supplement and should be shown. Further, to be able to claim that something is centrosomal, the centrosome must be shown - the authors would need to find a stable marker (see point 6). Alternatively, if authors would clearly show that the whole centrosome decomposes in AP-2 μ KO NPCs, the graph description in Fig. 3J should be changed to "Perinuclear MT asters" or something similar.

We thank the reviewer for this comment. We have re-done the EB3 experiment in WT and KO NPCs, where we have also marked the centrosome by γ -Tubulin-GFP expression. We were now able to visualize EB3 as MT plus end and observed only a minimal binding of EB3 along the length of MT. Our new analysis of EB3 track events (using TrackMate plugin in ImageJ) indicate that AP-2 KO NPCs reveal a significantly decreased amount of centrosomal EB3 tracks and increased amount of EB3 tracks originating further away from the centrosome (Fig. 3I-N).

We now re-considered the data with "static" EB3 phenotype and excluded them from our analyses, because we believe that these cells likely represent dying cells.

9. The experiments in MEFs could nicely complement the first part of the study, however, many

important complementary experiments are missing which is not comprehensible. Also the language in the text describing the results is not clear.

We have now added a series of new experiments in MEF cells and made a strong effort to adapt the language in the text, describing this section.

Analysis of γ -TuRC components and the centrosome integrity experiments suggested above should be done also on MEFs and iMEFs (always wt vs. AP-2 μ KO), including correctly performed MT regrowth experiments.

We have now performed a rigorous analysis of γ -TuRC components (GCP2 and GCP3) and centrosome morphology using co-immunostaining for CEP135 (as suggested by reviewer) together with γ -Tubulin in pMEFs and iMEFs. Additionally, we have also performed MT regrowth experiments in these cells. Importantly, we found that regrowth of centrosomal MTs is strongly reduced in both pMEFs and iMEFs (in line with the data we obtained in NPCs, please see above).

In addition, there is no quantification of the experiment shown in Fig. EV5E and it should be complemented by the same experiment done on "normal" MEFs (wt vs. KO).

The quantification of the data shown in Fig. EV5E was presented in Fig. 5M (now Fig. 6E). We complemented these data with experiments in primary MEFs (Fig. EV5D,E). These new data support the proliferation defect in KO pMEFs, consistent with the phenotype we reported in the original manuscript.

Further, analysis of centrosome maturation (a centrosome marker plus γ -tubulin) should be done on MEFs and iMEFs (always wt vs. KO) comparing interphase vs. G2/M (pHH3-positive) cells.

We have now performed a rigorous analysis of centrosome morphology using co-immunostaining for CEP135 (as suggested by reviewer) together with γ -Tubulin in all three cell types (NPCs, pMEFs and iMEFs). Our new data indicate that the centrosome number is abnormal in all cell types and support the original conclusion of loss of centrosome cycle integrity in cells lacking AP-2. Our analysis of centrosome number and microtubule nucleation was focused on cells in interphase; however, we also combined these analyses with the impact of these aberrations on mitotic fidelity.

Quantification of the Fig. 5J regarding cell number ["# of iMEFs (% of WT)] need to be shown.

The cell number of WT and AP-2 KO iMEFs is shown now in Fig. EV6A.

Cell cycle analysis (FACS) would be beneficial for both pMEFs and iMEFs (always wt vs. KO) as well.

We have now performed cell cycle analysis using flow cytometry in both WT and KO pMEFs and iMEFs. These data can be found in Fig. 5K, Fig. EV5C, Fig. EV6B,C. These

new data support the rescue of proliferation in KO iMEFs, the phenotype we reported in the original manuscript.

Minor points:

1. Evidence for AP-2 knock-out efficiency is strong by analogy and reference to the previous work of the authors. However, in a new system, it should also be shown in DIV3 tamoxifen-treated NPCs using the anti-AP-2 μ antibody (which is apparently working, since it was used for another experiment in Fig. EV2A).

We thank the reviewer for this comment. However, we believe that the fact that AP2M1 is significantly decreased in MS analysis of total proteome from these cells (please see AP2M1 in Fig. 1H, also specified in the text) already confirms the fact that both AP-2alpha and AP-2mu subunits are strongly decreased in our model. As mentioned above, it is very challenging to obtain lysates from AP-2 KO NPCs due to the cell death phenotype.

2. Line 46 -The sentence is ambiguous and needs to be rephrased. In our opinion, "The proliferative capacity of NPCs is determined..." mostly by cell-intrinsic factors which then dictate the number of division rounds and the varying identity of daughter cells. And not vice-versa.

We have now modified the sentence according to the reviewer's suggestion.

3. Line 55 - Centrosomes might actually be enclosed by a membrane, at least in *Caenorhabditis elegans* (<https://www.sciencedirect.com/science/article/abs/pii/S0960982222019960>).

We thank the reviewer for pointing this work to us. We have now included the reference by Maheshwari et al, 2023 in the paper and modified the sentence accordingly.

4. Line 61 - Accompany the citation of Schiebel, 2000, by newer reviews; taking into account for example newer findings on CKAP5 and γ -TuRC.

We thank the reviewer for this comment. We have now included the references by Alie et al, 2023, where γ TuRC recruitment and activation at interphase centrosomes has been shown to be controlled by the microtubule polymerase CKAP5 (ch-TOG), as well as the reviews by Liu et al., 2020 and Thawani & Petry, 2021. We also now cite the paper by Roostalu et al., 2015, where ch-TOG has been shown to cooperate with γ TuRC in chromatin- or augmin-mediated nucleation during spindle assembly.

5. Line 93-95 - The sentence and the citations are imprecise. Some of the cited works are about non-centrosome/spindle/midbody proteins regulating the mitosis actually through membrane morphology (e.g. Epsin1). Thus, the sentence should be rephrased (divide clearly canonical vs. non-canonical effects of CME components on the described processes and also distinguish clearly canonical and non-canonical/novel localization of the discussed CME components) and

add only relevant citations added. Some of the cited works are not linked to the topic directly (e.g. He et al, 2002); and this could be confusing for readers outside of the field.

We thank the reviewer for this comment. We followed the reviewer's suggestion and modified this section to contain the work referring to the membrane-trafficking – independent “non-canonical functions” of CME components in neuronal progenitors proliferation. Although, we would like to note that in the “endocytosis” field the role of Epsin1 in mitosis (even though this function is mediated by its role in membrane trafficking) is considered to be non-canonical. The canonical function of CME is generally the one occurring at the cell plasma membrane.

6. Use γ -tubulin instead of GCP1 (Figure EV2)- it is an uncommon term and could be mistakenly associated with another protein. Also the term "astral" microtubules is classically given to the MTs emanating for mitotic spindle poles (mitotic centrosomes) towards the cell membrane. It seems that the authors use this term also for interphase cells. We would recommend using the term "centrosomal microtubules" for MTs emanating from the centrosome in the interphase.

We thank the reviewer for this comment. We have replaced GCP1 with γ -tubulin. We have also adopted the term “centrosomal microtubules” throughout the manuscript.

7. Check for typos throughout the manuscript (AP2- μ instead of AP-2 μ in Fig. EV2A, Line 57 - animals cells, Line 297 - Fig. 5M, Fig. EV5E legend: iMEFs (not NPCs), Line 840 - missing spaces etc.).

We apologize for these typos that have been thoroughly corrected throughout the manuscript.

8. Show GCP3 and AP-2 μ colocalization in the Supplement (Missing pictures for the data in the graph in Fig. 2E)

The colocalization of GCP3 and AP-2 μ has been shown in Fig. EV2B (now Fig. EV2D).

9. Always use "neuronal progenitors" and not only "neuronal" when you are describing data acquired on NPCs. "Neuronal proliferation" - lines 277 and 279 is generally incorrect.

We thank the reviewer for this comment. We have adopted the term neuronal progenitor proliferation instead of neuronal proliferation throughout the manuscript.

10. Line 307 - "where it interacts with components..." This is not so straightforward. It is likely; however, the definite proof would be to see both γ -TuRC and AP-2 on isolated centrosomes or at least in a centrosomal fraction (without cytoplasmic complexes).

We thank the reviewer for this comment. We have modified the sentence accordingly to tone down the “interaction of AP-2 with γ -TuRC at centrosomes”. The sentences now read as follows: “We find that the AP-2 complex associates with components of the γ -TuRC complex and regulates centrosomal MT nucleation”.

11. Line 308 - MT anchoring was not addressed at all. MT nucleation is likely true based on the data, however, MT regrowth experiments and EB3 live imaging must be redone correctly (see above).

We thank the reviewer for this comment. We have modified the sentence accordingly (please see above). We have also performed the new set of MT regrowth and EB3 live imaging experiments to support our conclusion on centrosomal MT nucleation.

12. Line - 348 "centrosome overduplication" - There are no data shown to support this. The authors checked only γ -tubulin staining. The major phenotype was "centrosome loss" (Fig. 2M), meaning γ -tubulin double-dot centrosomal signal was lost. The multiple centrosomes observed in the rest of the cells could be γ -tubulin cytoplasmic puncta that can form under various conditions as shown by others before. However, the puncta are usually no centrosomes (they lack centrioles). More experiments would be needed to address this properly (i.e. always showing a reliable centriolar counterstain to distinguish multiple centrosomes vs. acentrosomal puncta of centrosomal proteins)

We thank the reviewer for this comment. To indicate that the phenotype we observed with γ -Tubulin corresponds to "centrosomes" *per se*, we have now performed a series of experiments, where we co-stained γ -Tubulin with the centriole marker CEP-135, as also suggested by the reviewer above. Our data indicate that the centrosomes positive for both γ -Tubulin with CEP-135 are increased in number in KO NPCs, pMEFs and iMEFs compared to controls. Thus, we believe that the term "centrosome amplification" is now supported by our data.

13. Line 366 - "neuronal embryonic" - better: embryonic neuronal progenitor cells

We have modified the sentence accordingly.

14. Line 368 - "of the γ -TuRC" - better: "of the active γ -TuRC"

We have modified the sentence accordingly.

15. Lines 377-378 - "... whereby AP-2 regulates centrosome function at the level of MT nucleation by recruiting components of the γ -TuRC complex." - Rephrasing needed. The data rather show that AP-2 is required for γ -TuRC stability, however, additional experiments, suggested above, are needed to establish this (missing WBs - wt vs. AP-2 KO). No data regarding γ -TuRC centrosomal recruitment through AP-2 shown.

We thank the reviewer for this comment. We have now rephrased the sentence, which reads as: "This finding, together with our observation that the AP-2 μ subunit traffics to the centrosome, suggest a model whereby AP-2 regulates centrosome function at the level of MT nucleation by associating with components of the γ -TuRC complex."

16. Lines 388-389 - "AP-2 may contribute to the expansion of pericentriolar material at the onset of mitosis by recruiting additional γ -TuRC complexes, thereby increasing MT nucleation activity as the cell cycle progresses." - Not shown, but easy to show by comparing interphase and mitotic centrosomes in wt vs. KO as suggested above

We agree with the reviewer that based on our data we cannot distinguish between the role of AP-2 in centrosome function in interphase versus mitosis and this is something we would like to address in the future by analyzing AP-2 localization at centrosomes in cells with synchronized cell cycle. The sentence quoted by the reviewer contained rather a hypothetical model rather than the statement based on our results. Nevertheless, we have rephrased it now to indicate that there are not enough data available at this stage to support this hypothesis. The sentence reads now as follows:

"We speculate that AP-2 may contribute to the expansion of pericentriolar material at the onset of mitosis by recruiting additional γ -TuRC complexes, thereby increasing MT nucleation activity as the cell cycle progresses; however more data are required to support this hypothesis."

17. Line 433 - use relative centrifugation force (rcf) rather than rpm

We have replaced rpm units with rcf.

18. Line 648 - typo - Pitstop2

We have thoroughly checked now for all typos in the manuscript.

19. Line 498 - "NPCs co-expressing (X)" - Unclear what does the X mean?

We apologize for this typo. We corrected the sentence accordingly.

20. Line 607 - "...by manually selecting the cell body using ImageJ selection tools (ROI)," - How was the cell boundary assessed? How was the background subtraction carried out? Were the imaging parameters kept constant among related experiments? This all should be stated. Immunofluorescence analysis should also focus directly on centrosomes as mentioned above.

We thank the reviewer for this comment. We have now added the following information to the section: Background subtraction was performed by selecting the ROI within the image field of view not containing cells. The cell boundary was defined using Vimentin co-immunostaining as a mask, and/or using phase contrast images of the cells. All imaging parameters were kept constant among related experiments.

Also, as stated above we performed new set of experiments where GCP2/GCP3 immunofluorescence analysis is focused on centrosomes.

21. The authors should consult a statistician for several analyses. For Fig. 1L,M,N,O, Fig. 2M, O, the t-test is not correct. A chi-square test might be more appropriate:

<https://www.sciencedirect.com/science/article/pii/S001216061001016X>

Figure EV3H,I - the data seem to be longitudinal, repeated measures ANOVA might be more appropriate.

We thank the reviewer for this comment. We now performed χ^2 square analysis for FACS cell cycle data, using original cell counts, as suggested in Xu et al., 2010 work provided by the reviewer. This analysis indicated that the distribution of cells among the cell cycle phases is significantly different between WT and KO condition in NPCs and pMEFs (this information is now provided in the figure legend). However, since χ^2 square doesn't allow to compare the differences between two conditions within categories itself (e.g. G1 versus S phase between WT and KO), we have additionally performed two-way ANOVA analysis for these data. Corresponding asterisks are now indicated in the figures.

Since chi-square calculations are only valid when all expected values are greater than 1.0 and at least 20% of the expected values are greater than 5, statistical differences among the groups data with "centrosomal phenotypes" were all analyzed now using two-way ANOVA. This is due to the fact that e.g. in WT condition "Multiple centrosomes" are often zero or close to zero in these datasets.

The data set EV3H,I (now Fig. EV3E-G) has also been analyzed using two-way ANOVA repeated measures.

22. Timer is missing in all movies and scale bars in EB3 movies. A starting slide describing the experiment would be good for all the movies. A centrosomal marker should be shown (even if the complete centrosome decomposition is confirmed in AP-2 KO, the control should be clear).

We have added now a starting slide, describing the experiment, as well as the timer and the scale bars. The EB3 live cell imaging has been now performed together with γ -Tubulin, as a marker of the centrosome.

November 16, 2023

RE: Life Science Alliance Manuscript #LSA-2023-02029-TR

Prof. Natalia L. Kononenko
Cologne Excellence Cluster on Cellular Stress Responses in Aging Associated Diseases
Faculty of Medicine and University Hospital Cologne, University of Cologne
Joseph-Stelzmann-Str. 26
Cologne, NRW 50931
Germany

Dear Dr. Kononenko,

Thank you for submitting your revised manuscript entitled "The AP-2 complex interacts with γ TuRC and regulates the proliferative capacity of neural progenitors". We would be happy to publish your paper in Life Science Alliance pending final revisions necessary to meet our formatting guidelines.

- please add ORCID ID for the corresponding author -- you should have received instructions on how to do so
- please add your supplementary figure, video, and table legends to the main manuscript text after the legends for the main figures
- please make sure the author order in your manuscript and our system match
- the full name (first, middle name as initials, last name) of each author should be given on the title page
- LSA allows supplementary figures, but no EV Figures; please update your callouts for the Supplementary Figures and Tables in the manuscript Fig EV1A=Fig S1A; Table EV1=Table S1; while supplementary figures use the system supplementary Fig S1 and Table S1; please do the same for videos
- please add callouts for Figures 6L; S6N and Video S2 to your main manuscript text
- please update the Data Availability statement to include the accession information for the Proteome data, which should be made publicly accessible at this point

Figure Checks:

- the blots in Fig. 2B (AP-2 α and γ -TUB) and Fig. S2A (AP-2 α and γ -TUB) look very similar to each other. please confirm that these are unique blots
- please provide the original blots used in Figure S2B, 8-week cortex as Source Data

A. FINAL FILES:

-- Summary blurb (enter in submission system): A short text summarizing in a single sentence the study (max. 200 characters including spaces). This text is used in conjunction with the titles of papers, hence should be informative and complementary to the title. It should describe the context and significance of the findings for a general readership; it should be written in the

present tense and refer to the work in the third person. Author names should not be mentioned.

B. MANUSCRIPT ORGANIZATION AND FORMATTING:

Sincerely,

Reviewer #1 (Comments to the Authors (Required)):

I am glad that the authors have addressed all my concerns experimentally. Now the endocytic and non-endocytic roles of AP-2 are more clearly delineated. The manuscript is now publishable.

Reviewer #2 (Comments to the Authors (Required)):

We do appreciate the perseverance and diligence of the authors during the revision and in answering all of our questions and comments. In our opinion, they did a superb job. We do not have any further comments. We gladly recommend the paper for publication in LSA and congratulate the authors on this exciting novel story.

November 27, 2023

RE: Life Science Alliance Manuscript #LSA-2023-02029-TRR

Prof. Natalia L. Kononenko
Center for Physiology and Pathophysiology
Faculty of Medicine, University Hospital Cologne, University of Cologne
Gleueler Straße 26
Cologne, NRW 50935
Germany

Dear Dr. Kononenko,

Thank you for submitting your Research Article entitled "The AP-2 complex interacts with γ TuRC and regulates the proliferative capacity of neural progenitors". It is a pleasure to let you know that your manuscript is now accepted for publication in Life Science Alliance. Congratulations on this interesting work.

DISTRIBUTION OF MATERIALS:

Again, congratulations on a very nice paper. I hope you found the review process to be constructive and are pleased with how the manuscript was handled editorially. We look forward to future exciting submissions from your lab.

Sincerely,
